# Neurosecretory protein GL stimulates food intake, de novo lipogenesis, and onset of obesity

Eiko Iwakoshi-Ukena[1,2†], Kenshiro Shikano[1†], Kunihiro Kondo[1], Shusuke Taniuchi[1], Megumi Furumitsu[1], Yuta Ochi[1], Tsutomu Sasaki[3], Shiki Okamoto[4,5,6], George E Bentley[2], Lance J Kriegsfeld[7], Yasuhiko Minokoshi[4,5], Kazuyoshi Ukena[1,7]*

[1]Section of Behavioral Sciences, Graduate School of Integrated Arts and Sciences, Hiroshima University, Higashi-Hiroshima, Japan; [2]Department of Integrative Biology and The Helen Wills Neuroscience Institute, University of California at Berkeley, Berkeley, United States; [3]Laboratory of Metabolic Signal, Institute for Molecular and Cellular Regulation, Gunma University, Maebashi, Japan; [4]Division of Endocrinology and Metabolism, Department of Homeostatic Regulation, National Institute for Physiological Sciences, Okazaki, Japan; [5]Department of Physiological Sciences, School of Life Science, Sokendai (The Graduate University for Advanced Studies), Hayama, Japan; [6]Second Department of Internal Medicine (Endocrinology, Diabetes and Metabolism, Hematology, Rheumatology), Graduate School of Medicine, University of the Ryukyus, Nakagami-gun, Japan; [7]Department of Psychology and The Helen Wills Neuroscience Institute, University of California at Berkeley, Berkeley, United States

*For correspondence: ukena@hiroshima-u.ac.jp

†These authors contributed equally to this work

Competing interests: The authors declare that no competing interests exist.

**Abstract** Mechanisms underlying the central regulation of food intake and fat accumulation are not fully understood. We found that neurosecretory protein GL (NPGL), a newly-identified neuropeptide, increased food intake and white adipose tissue (WAT) in rats. NPGL-precursor gene overexpression in the hypothalamus caused increases in food intake, WAT, body mass, and circulating insulin when fed a high calorie diet. Intracerebroventricular administration of NPGL induced de novo lipogenesis in WAT, increased insulin, and it selectively induced carbohydrate intake. Neutralizing antibody administration decreased the size of lipid droplets in WAT. *Npgl* mRNA expression was upregulated by fasting and low insulin levels. Additionally, NPGL-producing cells were responsive to insulin. These results point to NPGL as a novel neuronal regulator that drives food intake and fat deposition through de novo lipogenesis and acts to maintain steady-state fat level in concert with insulin. Dysregulation of NPGL may be a root cause of obesity.

## Introduction

Dysregulated energy balance can result in obesity and lead to serious health problems such as diabetes and cardiovascular disease (*Steinberger and Daniels, 2003*; *Hill et al., 2012*; *Ahima and Lazar, 2013*). Thus, it is important for human health to gain insight into the physiological mechanisms underlying the regulation of obesity. As obesity results mainly from overfeeding, most research to date has focused on hypothalamic regulation of feeding and satiety. Several hypothalamic neuropeptides and peripheral factors influence food intake and body mass (*Schwartz and Porte, 2005*; *Morton et al., 2006, 2014*). For instance, the arcuate nucleus (Arc) of the

**eLife digest** Throughout history, our ancestors needed to accumulate fat to survive during times when food sources were scarce. However, for most people in the modern age, food is abundant and eating too much is a major cause of weight gain, obesity and diseases affecting the metabolism. Obesity in particular, can lead to diseases such as diabetes and heart disease.

Hunger and appetite are regulated by proteins and other chemicals that act as messengers, for example insulin, and a region of the brain called the hypothalamus. However, the full mechanisms that regulate these sensations remain unclear. Only recently, a protein called NPGL was discovered in a part of the hypothalamus of birds and mammals. However, it was not known if NPGL plays a role in regulating eating habits and weight gain.

Iwakoshi-Ukena et al. have now discovered that NPGL is found in the hypothalamus of rats and is regulated by diet and insulin. When the gene for NPGL was manipulated to produce too much of the protein, rats fed a high calorie diet started to eat more, and gained more weight and body fat. Adding additional NPGL to their brains had the same effect. When the animals were fed a normal diet, NPGL only moderately affected how much they ate, but it substantially increased how much fat they produced. Iwakoshi-Ukena et al. also observed that when animals were starved and insulin levels were low, the rats started to produce more NPGL. These results suggest that NPGL plays a role in fat storage when energy sources are limited, and can contribute to obesity when too much NPGL is produced in animals on a high calorie diet.

These findings indicate that NPGL could be an additional brain chemical that regulates hunger and fat storage in mammals. A next step will be to reveal the specific mechanisms by which NPGL regulates overeating and fat accumulation. These findings will further advance the study and treatment of obesity and obesity-related diseases.

hypothalamus produces neuropeptide Y (NPY)/agouti-related peptide (AgRP) and α-melanocyte-stimulating hormone (α-MSH), which are potent orexigenic and anorexigenic factors, respectively (*Schwartz and Porte, 2005*; *Morton et al., 2006*, *2014*). In addition, leptin and insulin derived from peripheral tissues act on the hypothalamus to influence energy homeostasis (*Baskin et al., 1999*; *Varela and Horvath, 2012*). Precise hypothalamic control of energy balance affects puberty, thermoregulation, energy storage, survival, and other critical processes during different life-history stages (*Schneider et al., 2013*). Prior to the last decade, several bioactive peptides in the brain were discovered: neuromedin S, TLQP-21, nesfatin-1, and neuroendocrine regulatory peptide (NERP) (*Mori et al., 2005*; *Bartolomucci et al., 2006*; *Oh-I et al., 2006*; *Yamaguchi et al., 2007*). These neuropeptides are also involved in energy homeostasis and feeding behavior (*Ida et al., 2005*; *Bartolomucci et al., 2006*; *Oh-I et al., 2006*; *Toshinai et al., 2010*). Recently, nonadecaneuropeptide derived from Acyl-CoA binding domain-containing seven was found to be a novel anorexigenic factor in the mouse hypothalamus (*Lanfray et al., 2016*). Despite considerable progress in understanding the regulation of energy homeostasis over the last several decades, the neural control of hyperphagia or obesity is not completely understood. To further understand the mechanism regulating energy intake and/or storage, we sought to identify previously unknown bioactive substances in the hypothalamus that regulate energy metabolism.

As part of our search for novel neuropeptides and/or peptide hormone precursors in the hypothalamus, we identified a novel cDNA in the chicken hypothalamus and deduced a precursor protein including a secretory protein of 80 amino acids (*Ukena et al., 2014*). This small protein has Gly-Leu-$NH_2$ at its C-terminal and was named neurosecretory protein GL (NPGL). In chickens, subcutaneous infusion of NPGL increased body mass gain, suggesting that NPGL may be involved in growth processes, including energy homeostasis (*Ukena et al., 2014*). Subsequently, we found homologous *Npgl* genes in mammals, including human, rat, and mouse; the primary structure of NPGL is highly conserved among mammals and avian species (*Figure 1—figure supplement 1A*). Rat NPGL assumes a circular structure, although the mature structure has not been determined (*Figure 1A*). Given the effects of NPGL administration observed in chickens, along with the highly conserved nature of this gene across species, we hypothesized that NPGL and its precursor serve a prominent,

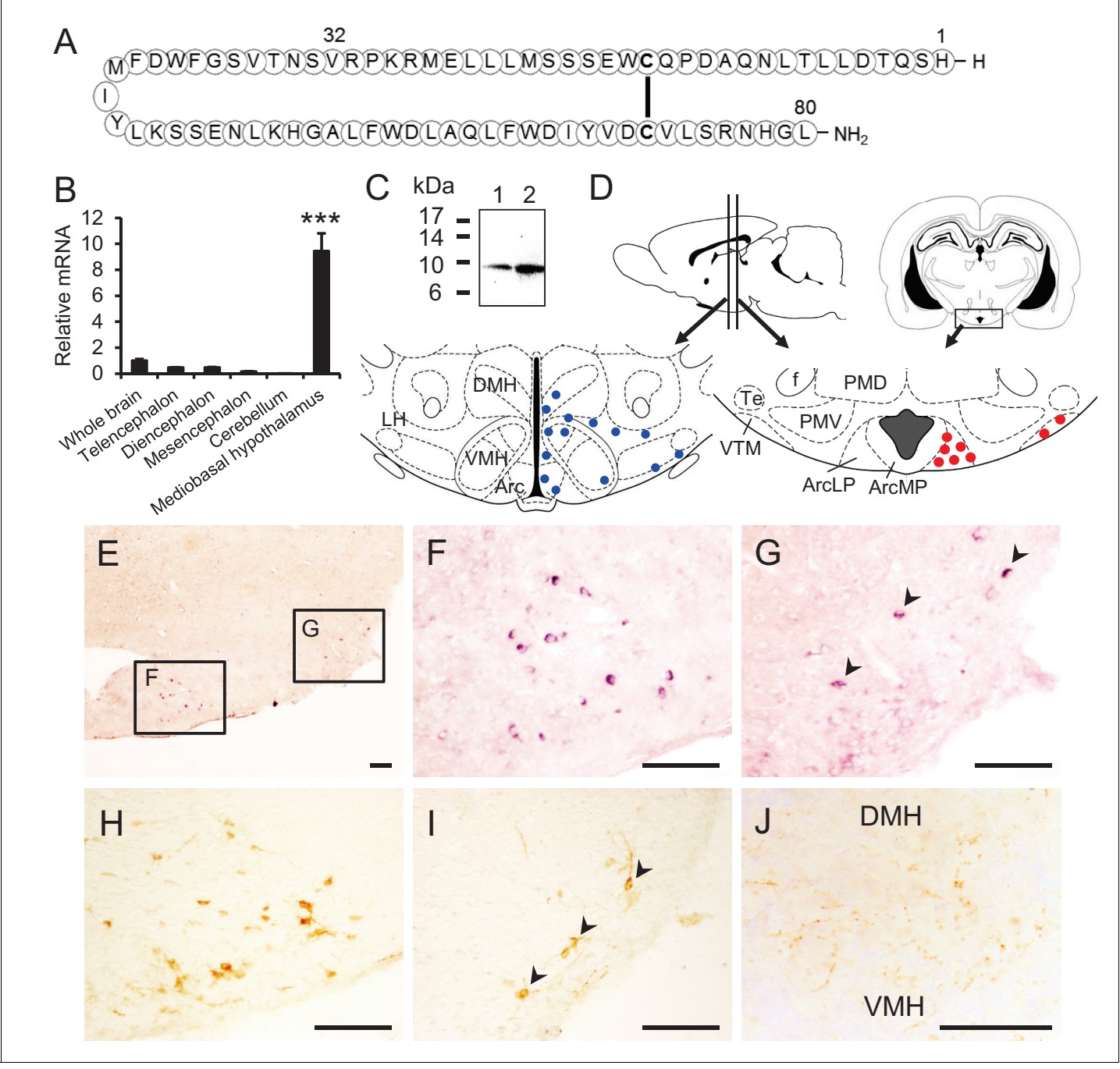

**Figure 1.** Structure of NPGL and expression of NPGL in rats. (**A**) The amino acid structure of NPGL is shown schematically. The bold line between cysteine residues indicates a disulfide bond. (**B**) Expression levels of the NPGL-precursor mRNA in the entire brain and different brain regions, including the telencephalon, diencephalon, mesencephalon, cerebellum, and mediobasal hypothalamus (n = 4). (**C**) Western blot analysis of mature NPGL in the hypothalamus. Synthetic NPGL served as a reference marker (1). The extract of the hypothalami from five rats (2). (**D**) Schematic representation of the localization of NPGL-immunoreactive fibers (blue dots) and cells (red dots) in the mediobasal hypothalamus. Abbreviations; Arc: the arcuate nucleus, ArcLP: lateroposterior part of the Arc, ArcMP: medial posterior part of the Arc, DMH: doromedial hypothalamus, f: fornix, LH: lateral hypothalamus, PMD: dorsal premammillary nucleus, PMV: ventral premammillary nucleus, Te: terete hypothalamic nucleus, VMH: ventromedial hypothalamus, and VTM: ventral tuberomammillary nucleus. (**E–G**) Photomicrographs of the cells containing NPGL-precursor mRNA in the mediobasal hypothalamus. The squares including the ArcLP and VTM are shown magnified in (**F**) and (**G**), respectively. Arrowheads in (**G**) indicate signals. Scale bar = 100 μm. (**H and I**) Photomicrographs of NPGL-immunoreactive cells in the ArcLP (**H**) and VTM (**I**). Arrowheads in (**I**) indicate signals. Scale bar = 100 μm. (**J**) Photomicrograph of NPGL-immunoreactive fibers between the DMH and VMH. Scale bar = 100 μm. Mean ± s.e.m. (one-way ANOVA with Tukey's test as a post-hoc test: ***p<0.005).

*Figure 1 continued on next page*

*Figure 1 continued*

The following figure supplement is available for figure 1:

**Figure supplement 1.** Amino acid sequences and expression site of NPGL.

unexplored role in energy homeostasis in mammals. More recently, we found that NPGL could induce food intake in mice (*Matsuura et al., 2017*). However, the physiological significance of NPGL in metabolic control in mammals remains to be elucidated.

The present investigation sought to characterize whether NPGL impacts food intake and energy metabolism using a rat model. To accomplish this goal, we first examined the pattern of expression in the brain and peripheral tissues, along with the specific localization and distribution of NPGL-producing cells in the brain. Subsequently, we investigated the biological action of NPGL and its precursor by overexpression of the precursor gene for *Npgl* in the hypothalamus, and intracerebroventricular (i.c.v.) infusion of NPGL or a specific antibody directed against this protein. We further examined the effects of NPGL on food intake, blood chemistry, and body composition when animals were fed normal chow, high calorie diet, or macronutrient diet. Finally, we determined that NPGL plays a role in monitoring energetic status and appropriate adjustment of feeding and energy metabolism.

## Results

### Expression of *Npgl* mRNA and NPGL in the hypothalamus

First, we examined the expression of *Npgl* mRNA in the brain and various peripheral tissues in human, rat, and mouse. The results revealed that *Npgl* mRNA expression is high in the brain and testis of human and rat, but in mouse it only exhibits high expression in the brain (*Figure 1—figure supplement 1B–D*).

To gain insight into the functional role of NPGL, we assessed the neuroanatomical localization of *Npgl* mRNA and its mature protein in the rat brain. *Npgl* mRNA was exclusively expressed in the mediobasal hypothalamus (*Figure 1B*). Western blot analysis demonstrated the presence of the mature form of NPGL in the hypothalamus (*Figure 1C*). Histological analyses showed that both *Npgl* mRNA and its mature protein were localized to the lateroposterior division of the Arc (ArcLP) and the ventral tuberomammillary nucleus (VTM) (*Figure 1D–I*). Immunoreactive cells were detected only after rats were treated with colchicine to prevent axonal transport of NPGL. NPGL-immunoreactive fibers were observed only in the hypothalamus (*Figure 1D and J*). This latter result suggests that NPGL's physiological role is mostly limited to regulation of the hypothalamus.

Because NPGL is mainly localized to the Arc, an important feeding and energy metabolic center, we speculated that NPGL produced in the ArcLP might be involved in the regulation of feeding behavior and body mass. Therefore, we investigated the effects of chronic exposure to NPGL on food intake, blood chemistry, and body composition in the following experiments.

### Effects of NPGL-precursor gene overexpression on food intake, body mass, adiposity, and blood insulin

The analysis of in vivo translation of the NPGL-precursor gene was conducted to survey the more long-term effect on phenotype, including body mass. To explore the effects of chronic NPGL-precursor gene (*Npgl*) overexpression in the mediobasal hypothalamus, we prepared an adeno-associated virus (AAV) that would allow chronic expression of the NPGL precursor protein (*Figure 2—figure supplement 1A*) and injected it into the hypothalamus of rats (*Figure 2—figure supplement 1B and C*). We then monitored food intake (high-fat/high-sucrose diet; high calorie diet, and normal chow) for 6 weeks (*Figure 2*). When we investigated the effect of *Npgl* overexpression in rats fed with high calorie diet, cumulative food intake, body mass, blood insulin, and WAT mass also markedly increased (*Figure 2A–D* and *Table 1A*).

In rats fed normal chow, daily food intake did not change until approximately 3 weeks after the AAV injection, but then slightly increased thereafter (*Figure 2—figure supplement 2A*). Overall,

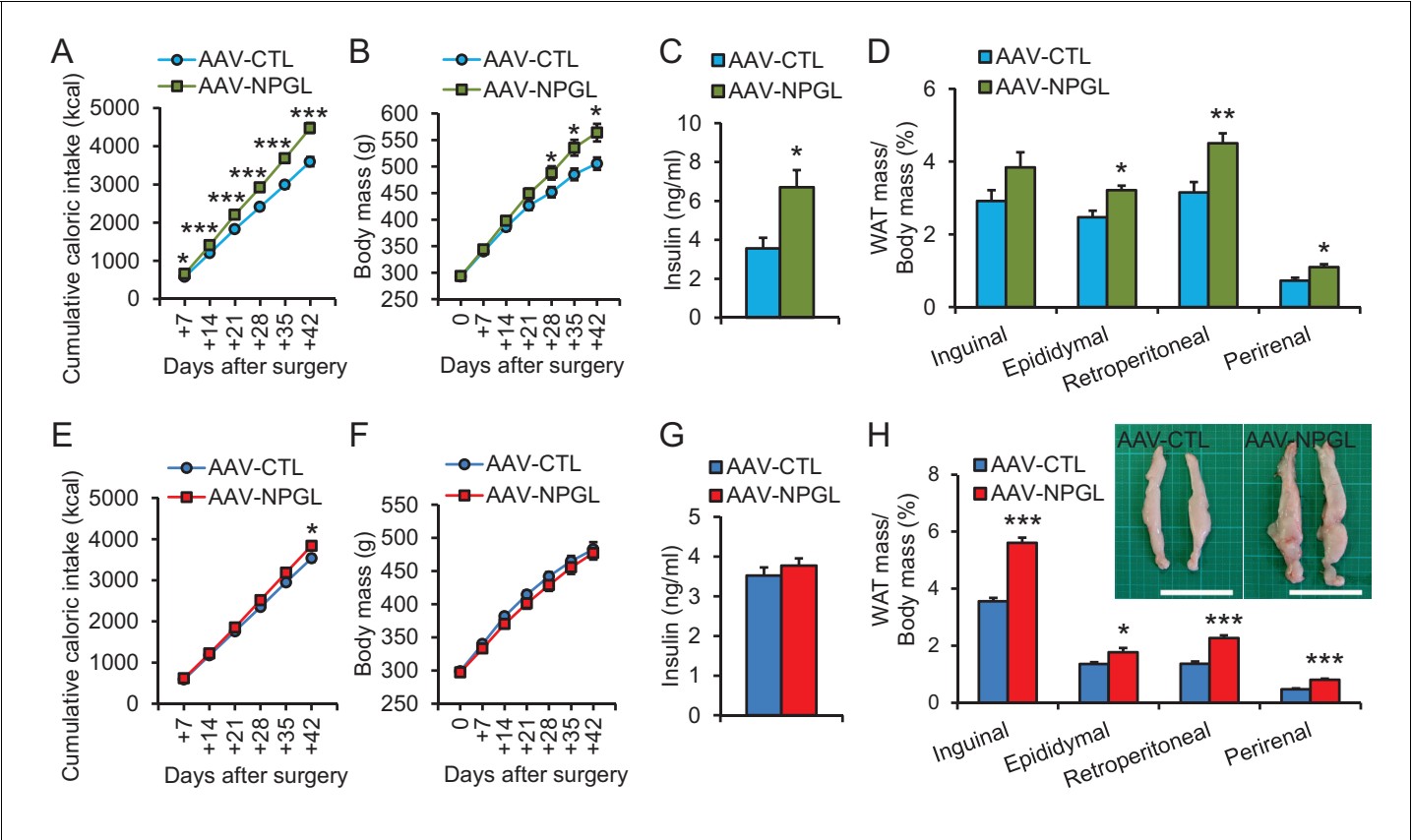

**Figure 2.** The effects of *Npgl* overexpression. The panels show the data obtained by the injection of AAV-based control vector (AAV-CTL) or AAV-based NPGL-precursor gene vector (AAV-NPGL) in high calorie diet (**A–D**) and normal chow (**E–H**). (**A**) Cumulative food intake (n = 8). (**B**) The body mass (n = 8). (**C**) Serum insulin levels (n = 6–7). (**D**) Ratios of inguinal, epididymal, retroperitoneal, and perirenal WAT mass/body mass (n = 6–7). (**E**) Cumulative food intake (n = 6–7). (**F**) The body mass (n = 6–7). (**G**) Serum insulin levels (n = 6–7). (**H**) Ratios of inguinal, epididymal, retroperitoneal, and perirenal WAT mass/body mass (n = 6–7) and representative photographs of retroperitoneal WAT. Scale bar = 5 cm. Mean ± s.e.m. (Student's *t*-test: *$p < 0.05$, **$p < 0.01$, ***$p < 0.005$).

The following figure supplements are available for figure 2:

**Figure supplement 1.** Construction of AAV-based vectors and verification of overexpression.

**Figure supplement 2.** The effects of *Npgl* overexpression in normal chow.

cumulative food intake significantly increased (*Figure 2E*). No overall effect on body mass was observed (*Figure 2F*). Despite no effect on body mass, *Npgl* overexpression caused a significant increase in the mass of WAT and the size of adipocytes (*Figure 2H* and *Figure 2—figure supplement 2B*). The masses of interscapular brown adipose tissue (BAT), liver, heart, and kidney remained unchanged (*Figure 2—figure supplement 2C and D*), while the masses of soleus and gastrocnemius muscles did not increase as much as controls, as did body length (*Figure 2—figure supplement 2E–G*). Blood leptin increased, but other blood measures, including insulin, did not change (*Table 1B* and *Figure 2G*).

## Effect of NPGL-precursor gene overexpression on metabolism and de novo lipogenesis

We measured the $O_2$ consumption ($VO_2$) and $CO_2$ production ($VCO_2$) at four weeks after AAV injection in normal chow-fed rats. There were no significant differences in the $VO_2$ and $VCO_2$ of treatment and control animals (*Figure 3A and B*). Although overall energy expenditure did not change, the respiratory quotient (RQ) was significantly elevated in the *Npgl* overexpression group

**Table 1.** Blood chemistry during *Npgl* overexpression.

**A. Blood chemistry during *Npgl* overexpression under high calorie diet.**

|  | AAV-CTL | AAV-NPGL |
|---|---|---|
| Glucose (mg/dl) | 144 ± 6.1 | 156 ± 6.2 |
| Free Fatty Acid (mEq/l) | 0.462 ± 0.038 | 0.460 ± 0.033 |
| Triglyceride (mg/dl) | 192 ± 12.7 | 184 ± 13.7 |
| Cholesterol (mg/dl) | 149 ± 9.6 | 127 ± 6.1† |
| Insulin (ng/ml) | 3.56 ± 0.77 | 6.71 ± 0.90* |

**B. Blood chemistry during *Npgl* overexpression under normal chow.**

|  | AAV-CTL | AAV-NPGL |
|---|---|---|
| Glucose (mg/dl) | 105 ± 2.6 | 108 ± 2.6 |
| Free Fatty Acid (mEq/l) | 0.703 ± 0.028 | 0.618 ± 0.031† |
| Triglyceride (mg/dl) | 202 ± 13.4 | 253 ± 23.9† |
| Cholesterol (mg/dl) | 80.8 ± 4.0 | 85.4 ± 2.1 |
| Insulin (ng/ml) | 3.52 ± 0.21 | 3.77 ± 0.16 |
| Leptin (ng/ml) | 12.0 ± 0.59 | 18.5 ± 1.60*** |
| Corticosterone (ng/ml) | 511 ± 2.1 | 512 ± 2.1 |

†$p <0.1$, *$p<0.05$, ***$p<0.005$.

(*Figure 3C*). The locomotor activity of the two groups did not differ (*Figure 3D*). Therefore, it is likely that the elevation of RQ value after *Npgl* overexpression is caused either by an upregulation of lipogenesis or downregulation of lipolysis in some tissues. To elucidate these possibilities, we analyzed the expression levels of lipogenic and lipolytic enzyme mRNAs in retroperitoneal WAT (rWAT) and liver. Specifically, we chose acetyl-CoA carboxylase (ACC), fatty acid synthase (FAS), stearoyl-CoA desaturase 1 (SCD1), glycerol-3-phosphate acyltransferase 1 (GPAT1), and adiponutrin (ADPN) as lipogenic enzymes, and carnitine palmitoyltransferase 1a (CPT1a), adipose triglyceride lipase (ATGL), and hormone-sensitive lipase (HSL) as lipolytic enzymes (*Shi and Burn, 2004*).

We found that mRNA expression levels of *Acc*, *Fas*, *Adpn*, and *Atgl* in rWAT significantly increased after *Npgl* overexpression, but no differences were detected in liver (*Figure 3E*). The protein level of FAS in rWAT also increased after *Npgl* overexpression, but the amount of phosphorylation of HSL was not different (*Figure 3F–H*). The fatty acid ratio of 16:1/16:0 significantly increased in rWAT of *Npgl* overexpression rats, but not 18:1/18:0, showing the activation of enzymatic activity of SCD1 (*Figure 3I and J*). These results indicate that the activation of NPGL-induced de novo lipogenesis occurs in WAT but not in liver.

## Effects of i.c.v. infusion of NPGL on food intake, adiposity, and de novo lipogenesis

In addition to the analysis in vivo translation of *Npgl* overexpression, we performed a 13 day chronic i.c.v. infusion of mature small protein, NPGL (15 nmol/day) using osmotic pumps in rats fed with high calorie diet and normal chow.

When we investigated the effects of chronic i.c.v. infusion of NPGL in rats fed with high calorie diet, food intake in the light period, blood insulin, cholesterol, and leptin all increased without changes in overall body mass (*Figure 4A–C* and *Table 2A*). Increases in the masses of inguinal WAT (iWAT) and perirenal WAT were also observed under this diet (*Figure 4D*).

Under normal chow, although no changes in food intake, blood chemistry, including insulin, or body mass were observed (*Figure 4E–G* and *Table 2B*), the masses of rWAT and perirenal WAT significantly increased (*Figure 4H*). The induction of de novo lipogenesis in rWAT but not liver was observed (*Figure 4—figure supplement 1*). In addition, acute i.c.v. injection of NPGL also induced de novo lipogenesis in rWAT (*Figure 4—figure supplement 2*). Analysis of the structure-activity

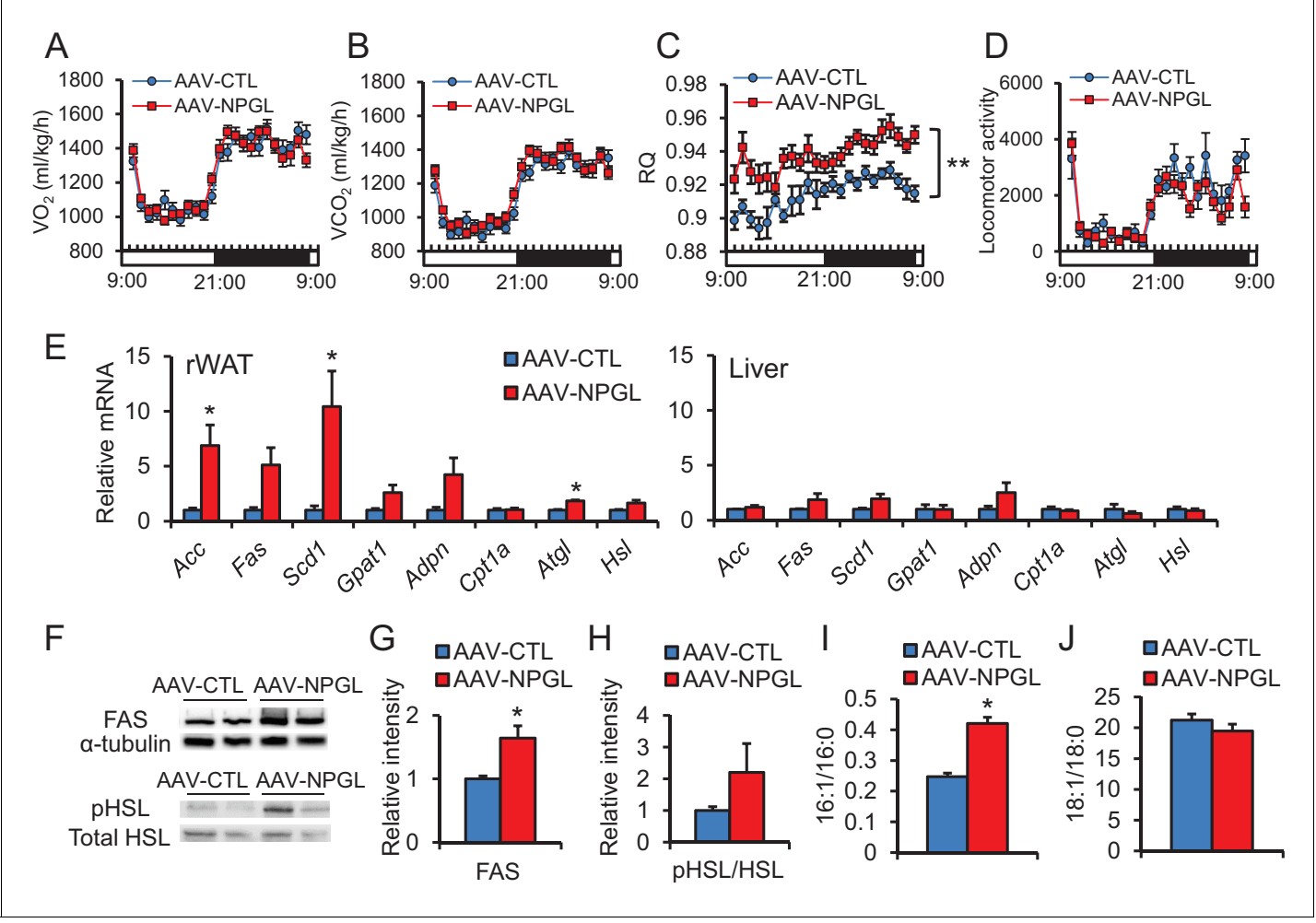

**Figure 3.** The effects of *Npgl* overexpression on $O_2/CO_2$ metabolism and lipogenic and lipolytic enzymes. All panels show data obtained by the injection of AAV-based control vector (AAV-CTL) or AAV-based NPGL-precursor gene vector (AAV-NPGL) in normal chow. (**A**) $O_2$ consumption ($VO_2$) measured in the metabolic cage (n = 6–7). (**B**) $CO_2$ production ($VCO_2$) measured in the metabolic cage (n = 6–7). (**C**) The respiratory quotient (RQ) measured in the metabolic cage (n = 6–7). (**D**) The spontaneous locomotor activity measured by infrared ray passive sensor (n = 6–7). (**E**) mRNA expression levels for lipogenic (*Acc*, *Fas*, *Scd1*, *Gpat1*, and *Adpn*) and lipolytic (*Cpt1a*, *Atgl*, and *Hsl*) enzymes in retroperitoneal WAT (rWAT) and liver (n = 5–7). Representative photographs (**F**) of the western blot and protein expression levels of FAS (**G**) and phosphorylated HSL (pHSL) (**H**) in retroperitoneal WAT (n = 6). (**I**) Ratio of fatty acids (16:1 and 16:0) in retroperitoneal WAT (n = 6–7). (**J**) Ratio of fatty acids (18:1 and 18:0) in retroperitoneal WAT (n = 6–7). Mean ± s.e.m. (two-way ANOVA followed by Bonferroni's test or Student's *t*-test: *p<0.05, **p<0.01).

relationship using a C-terminal Gly extended form of NPGL (NPGL-Gly; *Figure 4—figure supplement 3A*) and an N-terminal deletion form of NPGL [NPGL(32-80); *Figure 4—figure supplement 3A*] revealed that the longer form containing a disulfide bond is functional and C-terminal amidation is not necessary for induction of adiposity (*Figure 4—figure supplement 3B and C*). Furthermore, we measured the size of adipocytes in rWAT in rats fed normal chow. Infusion of 7.5 and 15 nmol/day NPGL increased the frequency of medium (3001–4000 µm$^2$) and large (>5001 µm$^2$) adipocytes, respectively, relative to vehicle controls, indicating a dose-dependent effect of NPGL (*Figure 4I*).

## Effect of i.c.v. infusion of antibody against NPGL on lipid droplets of WAT

To suppress the activity of endogenous NPGL, we performed a 13 day i.c.v. infusion of an antibody directed against NPGL in rats fed high calorie diet. Although food intake and body mass did not change (*Figure 4—figure supplement 4A and B*), blood cholesterol significantly decreased and the

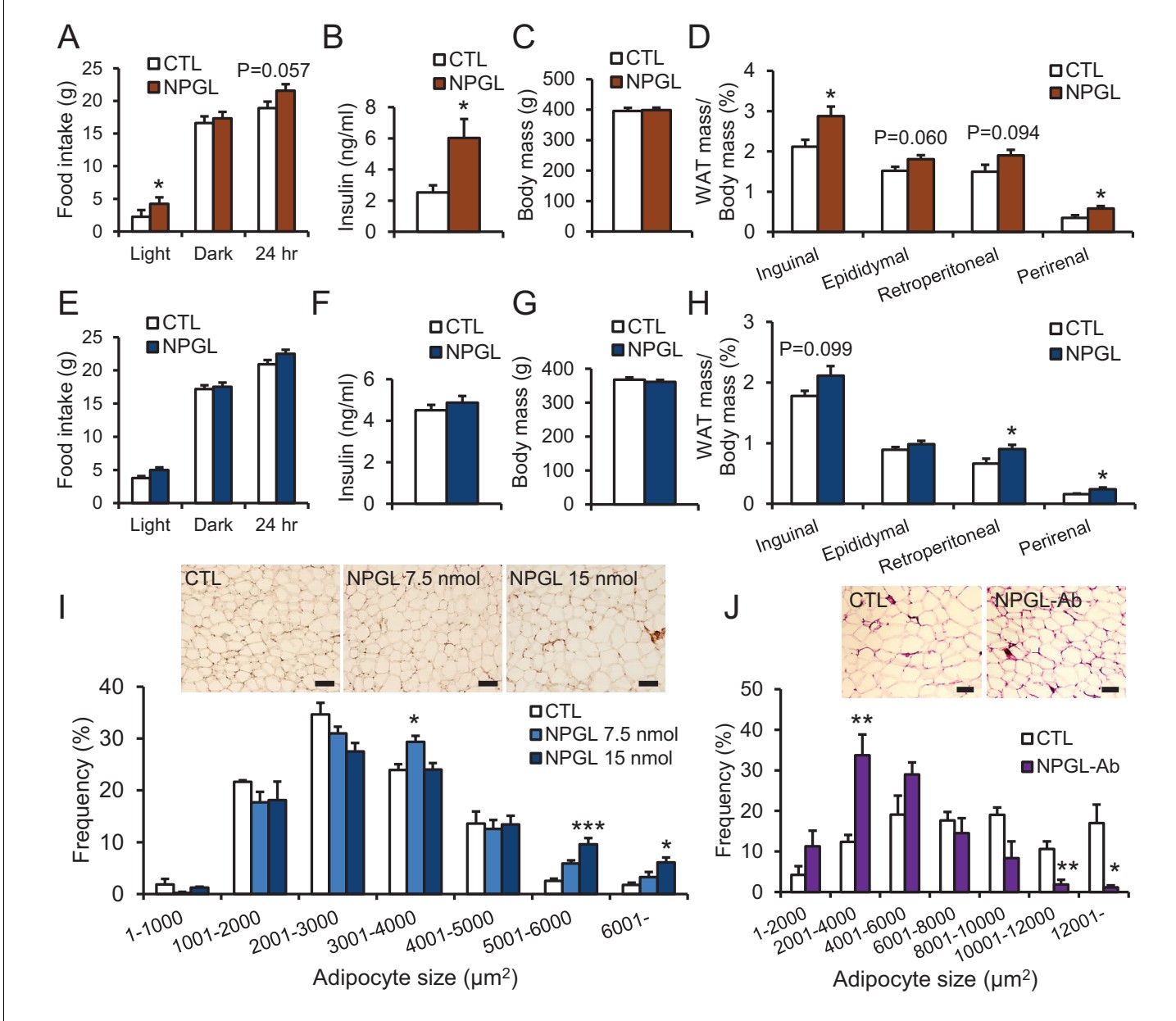

**Figure 4.** The effects of chronic i.c.v. infusion of NPGL or antibody against NPGL. All panels show data obtained by the infusion of vehicle (CTL) or NPGL (A–I) in high calorie diet (A–D) and normal chow (E–I), and the control IgG (CTL) or the antibody against NPGL (NPGL-Ab) in high calorie diet (J). (A) The average of food intake during the light period, dark period, or over 24 hr (n = 8). (B) Serum insulin levels (n = 8). (C) Body mass (n = 8). (D) Ratios of inguinal, epididymal, retroperitoneal, and perirenal WAT mass/body mass (n = 8). (E) The average of food intake during the light period, dark period, or over 24 hr (n = 7). (F) Serum insulin levels (n = 7). (G) Body mass (n = 7). (H) Ratios of inguinal, epididymal, retroperitoneal, and perirenal WAT mass/body mass (n = 7). (I) The frequency of various adipocyte sizes measured in 1000 $\mu m^2$ areas and representative photographs in sections of retroperitoneal WAT after the infusion of vehicle (CTL) or NPGL in normal chow (n = 4–5). Scale bar = 100 $\mu m$. (J) The frequency of various adipocyte sizes measured in 2000 $\mu m^2$ areas and representative photographs in sections of retroperitoneal WAT after the infusion of the control IgG (CTL) or antibody against NPGL (NPGL-Ab) in high calorie diet (n = 4). Scale bar = 100 $\mu m$. Mean ± s.e.m. (Student's *t*-test or one-way ANOVA with Tukey's test as a post-hoc test: *p<0.05, **p<0.01, ***p<0.005).

The following figure supplements are available for figure 4:

**Figure supplement 1.** The effects of chronic i.c.v. infusion of NPGL on lipogenic and lipolytic enzymes.

**Figure supplement 2.** The effects of acute i.c.v. injection of NPGL on lipogenic and lipolytic enzymes.

*Figure 4 continued on next page*

*Figure 4 continued*

**Figure supplement 3.** The effects of chronic i.c.v. infusion of synthetic analogs in normal chow.

**Figure supplement 4.** The effects of chronic i.c.v. infusion of antibody against NPGL in high calorie diet.

**Table 2.** Blood chemistry during chronic i.c.v. infusion of NPGL or the antibody against NPGL.

**A. Blood chemistry during chronic i.c.v. infusion of NPGL under high calorie diet.**

|  | CTL | NPGL |
| --- | --- | --- |
| Glucose (mg/dl) | 138 ± 5.7 | 135 ± 2.5 |
| Free Fatty Acid (mEq/l) | 0.526 ± 0.043 | 0.56 ± 0.066 |
| Triglyceride (mg/dl) | 101 ± 13.2 | 149 ± 20.9$^\dagger$ |
| Cholesterol (mg/dl) | 99.0 ± 5.1 | 120.4 ± 5.3* |
| Insulin (ng/ml) | 2.53 ± 0.45 | 6.03 ± 1.22* |
| Leptin (ng/ml) | 8.76 ± 1.0 | 17.07 ± 1.6*** |

**B. Blood chemistry during chronic i.c.v. infusion of NPGL under normal chow.**

|  | CTL | NPGL |
| --- | --- | --- |
| Glucose (mg/dl) | 123 ± 7.0 | 128 ± 7.2 |
| Free Fatty Acid (mEq/l) | 0.276 ± 0.017 | 0.289 ± 0.017 |
| Triglyceride (mg/dl) | 73.0 ± 4.5 | 101.7 ± 11.6$^\dagger$ |
| Cholesterol (mg/dl) | 66.5 ± 2.1 | 67.5 ± 3.4 |
| Insulin (ng/ml) | 4.51 ± 0.26 | 4.87 ± 0.33 |
| Leptin (ng/ml) | 8.84 ± 1.4 | 7.93 ± 0.70 |
| Corticosterone (ng/ml) | 523 ± 3.6 | 519 ± 1.4 |

**C. Blood chemistry during chronic i.c.v. infusion of the antibody against NPGL under high calorie diet.**

|  | CTL | NPGL-Ab |
| --- | --- | --- |
| Glucose (mg/dl) | 134 ± 2.38 | 137 ± 1.84 |
| Free Fatty Acid (mEq/l) | 0.529 ± 0.078 | 0.479 ± 0.048 |
| Triglyceride (mg/dl) | 129 ± 14.6 | 118 ± 18.8 |
| Cholesterol (mg/dl) | 111.9 ± 3.7 | 98.5 ± 4.7* |
| Insulin (ng/ml) | 2.23 ± 0.29 | 2.53 ± 0.33 |
| Leptin (ng/ml) | 9.14 ± 0.85 | 7.24 ± 0.51$^\dagger$ |

**D. Blood chemistry during chronic i.c.v. infusion of NPGL under macronutrient diet.**

|  | CTL | NPGL |
| --- | --- | --- |
| Glucose (mg/dl) | 116 ± 4.4 | 124 ± 4.4 |
| Free Fatty Acid (mEq/l) | 0.359 ± 0.034 | 0.416 ± 0.042 |
| Triglyceride (mg/dl) | 74.4 ± 6.4 | 97.7 ± 7.8* |
| Cholesterol (mg/dl) | 99.7 ± 3.5 | 90.8 ± 5.5 |
| Insulin (ng/ml) | 1.94 ± 0.16 | 3.05 ± 0.68 |
| Leptin (ng/ml) | 3.92 ± 0.32 | 6.24 ± 0.33*** |

$^\dagger$p <0.1, *p<0.05, ***p<0.005.

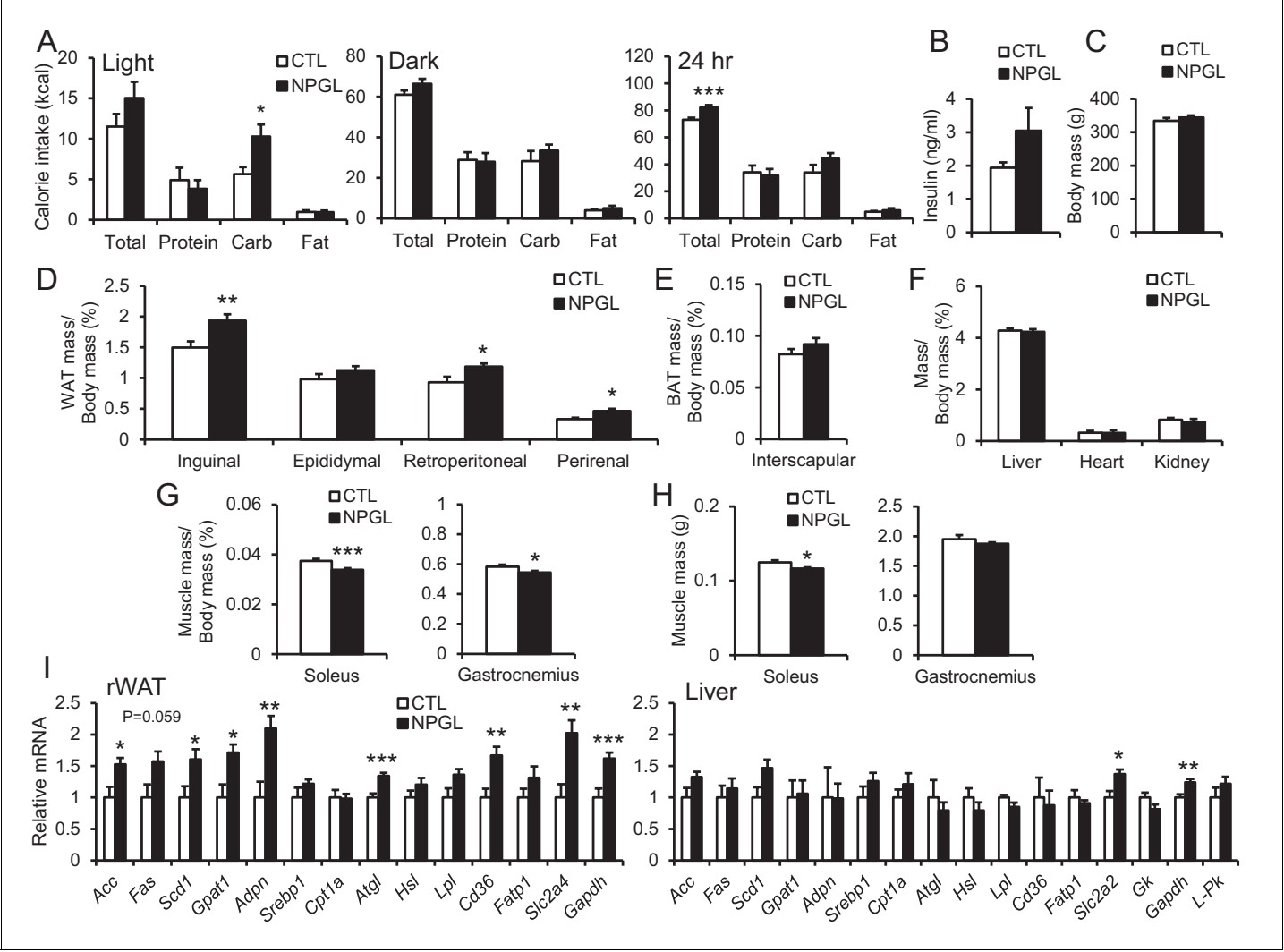

**Figure 5.** Feeding behavior of macronutrient diets during chronic i.c.v. infusion of NPGL. (**A**) The calorie intake of macronutrient diets (total, protein, carbohydrate, and fat) by the infusion of vehicle (CTL) or NPGL during the light period, dark period, or over 24 hr (n = 8). (**B**) Serum insulin levels (n = 8). (**C**) Body mass (n = 8). (**D**) Ratios of inguinal, epididymal, retroperitoneal, and perirenal WAT mass/body mass (n = 8). (**E**) Ratio of the interscapular BAT mass/body mass (n = 8). (**F**) Ratios of liver, heart, and kidney mass/body mass (n = 8). (**G**) Ratios of soleus and gastrocnemius muscle mass/body mass (n = 8). (**H**) Masses of soleus and gastrocnemius muscles (n = 8). (**I**) mRNA expression levels for lipogenic (*Acc*, *Fas*, *Scd1*, *Gpat1*, and *Adpn*) and lipolytic (*Cpt1a*, *Atgl*, and *Hsl*) enzymes, *sterol regulatory element binding protein 1* (*Srebp1*), *lipoprotein lipase* (*Lpl*), *Cd36*, *fatty acid transport protein 1 (Fatp1)*, *Slc2a4*, *Gapdh*, *Slc2a2*, *glucokinase* (*Gk*), and *liver pyruvate kinase* (L–Pk) in retroperitoneal WAT (rWAT) and liver (n = 7–8). Mean ± s.e.m. (Student's t-test: *p<0.05, **p<0.01, ***p<0.005).

The following figure supplement is available for figure 5:

**Figure supplement 1.** The effects of chronic i.c.v. infusion of NPGL under macronutrient diets.

mass of iWAT and blood leptin tended to decrease (*Table 2C* and *Figure 4—figure supplement 4C*). The lipid droplets in rWAT were significantly smaller after antibody infusion than those of the controls (*Figure 4J*). This result suggests that endogenous NPGL promotes adiposity, a finding in agreement with the results observed with NPGL infusion and *Npgl* overexpression.

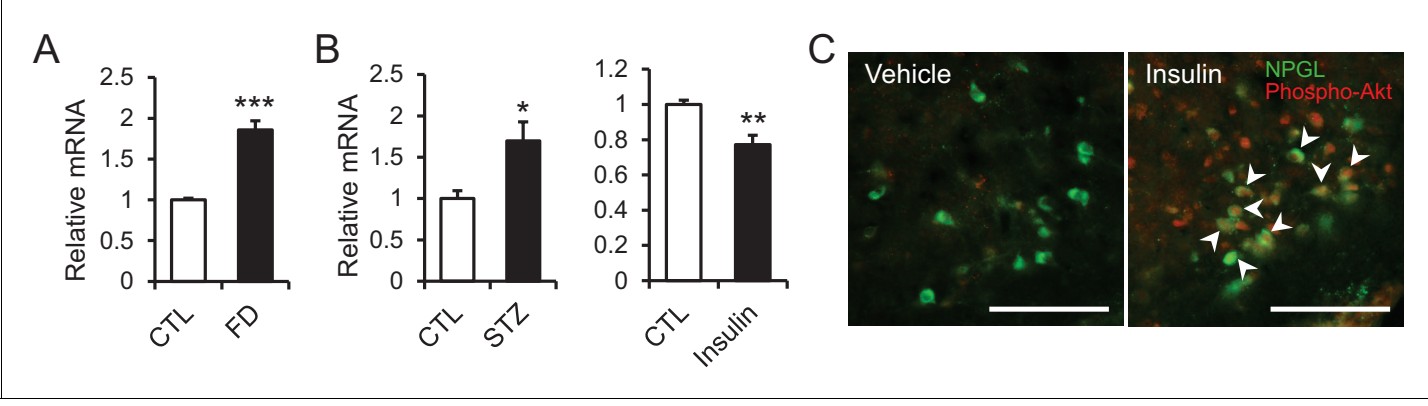

**Figure 6.** The changes of *Npgl* mRNA by fasting, streptozotocin or insulin and responsibility of NPGL-producing cells to insulin. (**A**) The effects of fasting. *Npgl* mRNA levels in the mediobasal hypothalamus of fasted rats (n = 8). (**B**) The effects of streptozotocin (STZ) or insulin. *Npgl* mRNA levels in the mediobasal hypothalamus after 7 days of i.p. injection of STZ (n = 7) or after 7 hr of i.p. injection of insulin (n = 6–7). (**C**) Induction of phosphorylated Akt (red) in NPGL-immunoreactive cells (green) after 45 min of i.c.v. injection of insulin in overnight fasted rats. Arrowheads indicate double-labeled cells (co-localization of phospho-Akt and NPGL). Scale bar = 100 µm. Mean ± s.e.m. (Student's *t*-test: *p<0.05, **p<0.01, ***p<0.005).

## Effects of NPGL on selective carbohydrate intake, adiposity, blood insulin, and de novo lipogenesis

As mentioned above, we observed that NPGL induced more potent food intake in animals fed a high calorie diet than those fed normal chow after *Npgl* overexpression and i.c.v. infusion of NPGL. Blood insulin increased only under a high calorie diet in both studies of *Npgl* overexpression and NPGL infusion. This high calorie diet includes high levels of sucrose as the carbohydrate source. Therefore, we hypothesized that NPGL may induce carbohydrate intake for the purpose of de novo lipogenesis and increase blood insulin levels. We tested this hypothesis using *ad libitum* selective feeding of macronutrient diets, protein, carbohydrate, and fat diets in the next series of experiments.

Rats infused with NPGL increased their intake of carbohydrate diet during the light period (***Figure 5A*** and ***Figure 5—figure supplement 1***). This carbohydrate feeding may result in increased total calorie intake over 24 hr (***Figure 5A*** and ***Figure 5—figure supplement 1***). Blood triglyceride and leptin increased without changes in insulin and overall body mass (***Figure 5B and C***, and ***Table 2D***). The masses of WAT also significantly increased in this experiment (***Figure 5D***). Although the masses of BAT, liver, heart, and kidney remained unchanged, the mass of the soleus muscle was lower than in controls (***Figure 5E–H***). In addition, mRNA expression of lipogenic enzymes increased in rWAT, but not liver (***Figure 5I***). Furthermore, mRNA expression of the fatty acid transporter (*Cd36*), glucose transporters (*Slc2a2* or *Slc2a4*), and a carbohydrate metabolism gene (*glyceraldehyde-3-phosphate dehydrogenase; Gapdh*) were upregulated in rWAT or liver (***Figure 5I***).

## Response of NPGL to fasting and insulin

To explore the role of NPGL in monitoring energetic state, the expression of *Npgl* mRNA was examined during a negative energy balance created through fasting. The expression of *Npgl* mRNA increased in rats fasted for 48 hr, with animals showing low blood glucose and insulin levels (***Figure 6A***). In addition, to manipulate blood glucose and insulin levels, we examined the expression of *Npgl* mRNA in a diabetic rat model in which animals possess high blood glucose and low insulin following i.p. streptozotocin (STZ; toxic to the beta cells of pancreas) administration. The expression of *Npgl* mRNA increased in STZ-treated rats (***Figure 6B***). In contrast, i.p. injection of insulin reduced the expression of *Npgl* mRNA (***Figure 6B***). These results indicate that *Npgl* mRNA expression is upregulated by fasting and low insulin and downregulated by high insulin.

In the final experiment, whether NPGL-producing cells respond to insulin was investigated. I.c.v. injection of insulin immediately induced phosphorylation of Akt, an insulin-response kinase, in NPGL cells (***Figure 6C***).

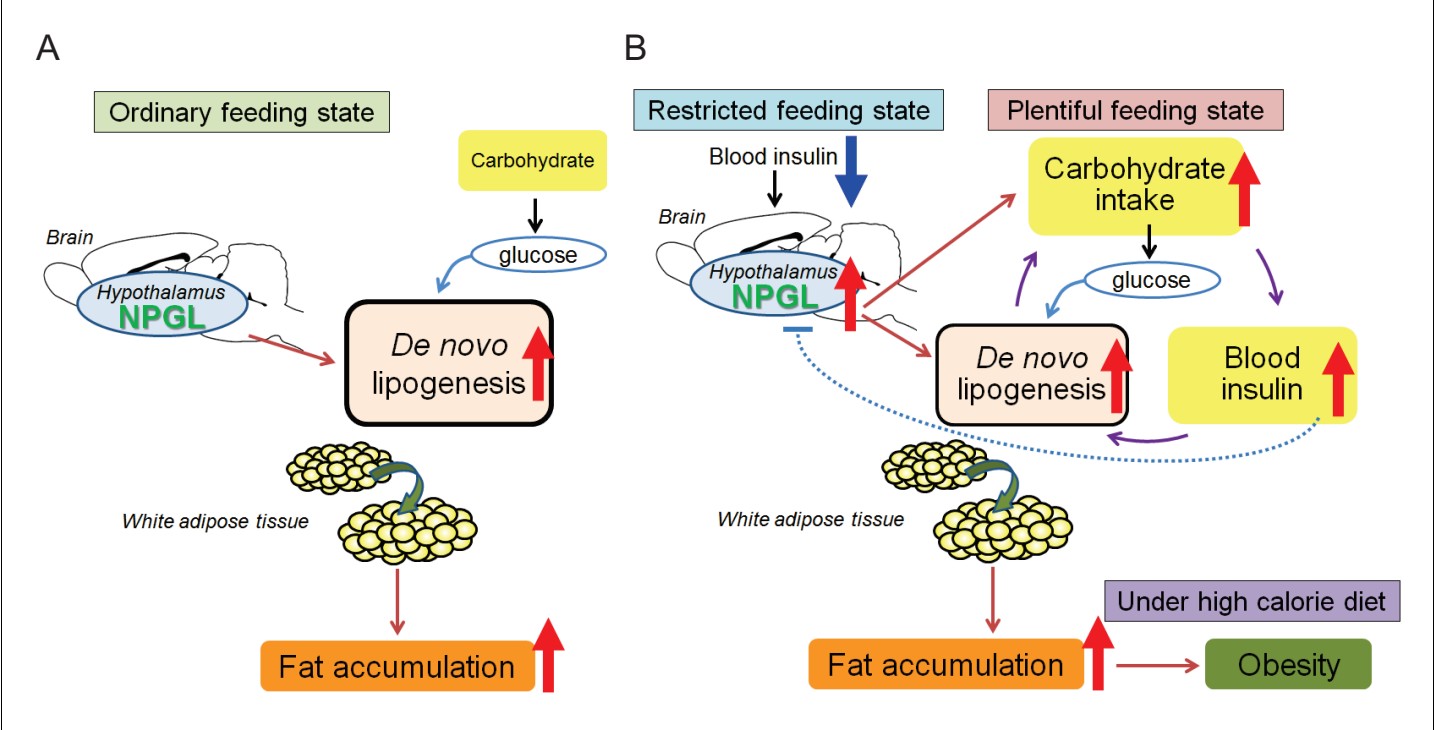

**Figure 7.** Potential role of NPGL in food intake and fat accumulation. Downward blue and upward red arrows indicate 'inhibition' and 'stimulation', respectively and other-colored arrows demonstrate downstream events. (**A**) During the ordinary feeding state, NPGL stimulates de novo lipogenesis and promotes fat accumulation of WAT. (**B**) In the restricted feeding state like fasting, a decrease in circulating insulin induces the expression of *Npgl* mRNA and NPGL production in the hypothalamus. When food is readily available, NPGL stimulates carbohydrate intake and induces de novo lipogenesis using available carbohydrate. Subsequently, NPGL stimulates more carbohydrate usage to provide the glucose substrate for lipogenesis. Finally carbohydrate intake increases blood insulin levels. This step can be repeated to achieve fat accumulation in WAT efficiently. Excess de novo lipogenesis by NPGL is inhibited through negative feedback actions of insulin to maintain steady-state fat storage in WAT. Under high calorie diet, NPGL induces overweight and finally obesity.

## Discussion

In the present study, we detected high expression of *Npgl* mRNA in the human, rat, and mouse brain, particularly in the mediobasal hypothalamus. Furthermore, we found that mature NPGL is produced in restricted hypothalamic nuclei and specifically plays a major role in food intake and fat accumulation in rats. Obesity is a burgeoning health issue worldwide, and yet how the brain regulates hyperphagia or adiposity is unclear. Under a high calorie diet including high sucrose, NPGL stimulates feeding behavior and increases blood insulin. At this time, blood glucose levels are not different in control and NPGL-treated groups, suggesting that insulin resistance is not induced by NPGL. Therefore, it seems that NPGL stimulates carbohydrate intake and induces de novo lipogenesis because these phenomena occur readily under high carbohydrate conditions (*Figure 7B*). Subsequently, NPGL stimulates more carbohydrate usage to provide the glucose substrate for lipogenesis, and finally accelerated carbohydrate intake increases blood insulin levels as an indirect action of NPGL (*Figure 7B*). This step can recur to achieve fat accumulation in WAT efficiently (*Figure 7B*). It is well known that insulin is an anabolic hormone, including its role in fat deposition in adipocytes (*Dimitriadis et al., 2011*). However, as revealed in the present experiments, administration of insulin inhibits the expression of *Npgl* mRNA (*Figure 6B*). This result suggests that excess de novo lipogenesis by NPGL may be inhibited through negative feedback action of insulin (*Figure 7B*). Together, the present findings point to NPGL as a novel neuronal lipogenic factor that maintains steady levels of fat storage in harmony with insulin. However, the excess and prolonged action of NPGL under high calorie diet finally leads to obesity (*Figure 7B*). In developing or developed countries, abdominal adiposity easily leads to obesity, which is associated with an increased risk of

metabolic syndrome and cardiovascular disease (*Oliveros et al., 2014*). The relationship between the action of human NPGL and its pathophysiological status has yet to be assessed, but the conserved nature of NPGL indicates that it is likely to play an important metabolic role in humans.

The present findings indicate that NPGL induces adiposity in WAT without remarkable changes in overall body mass in the experiments except for *Npgl* overexpression under high calorie diet. This increased fat accumulation, combined with no increase in muscle mass and body length, appears to explain the overall lack of body mass difference between the control and NPGL-treatment groups (*Figure 2—figure supplement 2E–G* and *Figure 5G and H*). Furthermore, this study shows that NPGL-mediated fat accumulation principally results from de novo lipogenesis in WAT but not in liver even when animals are fed normal chow and increases in food intake and blood insulin levels do not occur (*Figure 7A*). Indeed, the data from acute i.c.v. injection of NPGL show that NPGL can induce de novo lipogenesis in WAT rapidly (*Figure 4—figure supplement 2*). To our knowledge, our study is the first report on the presence of an endogenous hypothalamic mediator whose function is to control peripheral de novo lipogenesis independently of feeding behavior in animals. De novo lipogenesis is the biochemical process of converting non-lipid precursors into fatty acids for storage as energy (*Moore et al., 2014*). Recent evidence suggests that de novo lipogenesis in WAT plays an important role in maintaining metabolic homeostasis, although de novo lipogenesis in non-adipose tissues including liver leads to ectopic lipid accumulation, lipotoxicity, and metabolic stress (*Lodhi et al., 2011*; *Solinas et al., 2015*). In adipose tissues, glucose taken up via SLC2A4 (referred to as GLUT4) is converted into citrate via the glycolysis system and TCA cycle (*Rui, 2014*; *TeSlaa and Teitell, 2014*). Citrate is converted to palmitoleic acid by lipogenic enzymes including ACC, FAS and SCD1, and eventually stored as triglycerides via GPAT1 and ADPN (*Shi and Burn, 2004*; *Rui, 2014*; *Jeong et al., 2011*). In addition to lipogenic enzymes, the mRNA expression of *Atgl*, a lipolytic enzyme, also increased in NPGL-treated rats in the cases of three different experiments of this study (*Figures 3E* and *5I*, and *Figure 4—figure supplement 1A*). Furthermore, the mRNA expression of *Cd36*, a fatty acid transporter, increased in WAT of NPGL-infused rats under macronutrient diets (*Figure 5I*). Analyses using adipose tissue specific *Atgl* knockout mice and whole body *Cd36* knockout mice suggest that the fat oxidation system and fatty acid transport are required for de novo lipogenesis in adipose tissues and liver (*Mottillo et al., 2014*; *Clugston et al., 2014*). This result suggests that the activation of de novo lipogenesis by NPGL may be coupled with lipolysis to accelerate fat accumulation in WAT.

In contrast with its action during a positive energy state, NPGL may act to prepare animals for a negative energy state during food restriction, as fasting induced *Npgl* mRNA expression (*Figure 6A*). It is well known that the expression of anabolic neuropeptides such as NPY and AgRP are upregulated under negative energy balance (*Morton et al., 2006*). NPGL selectively stimulates carbohydrate intake during the light period when rats are not typically active or feeding (*Figure 5A*). Circulating insulin levels are lower at this time than during the dark period (*Marcheva et al., 2010*). Taken together, our data suggest that a decrease in circulating insulin during a restricted feeding state such as fasting may induce the expression of *Npgl* mRNA (*Figure 7B*). This mechanism of NPGL induction may serve as a hunger signal to prepare for fat accumulation during a food available state.

NPGL-producing cells are located in the ArcLP and the VTM in the posterior hypothalamus. In addition, we detected NPGL-immunoreactive fibers in the anterior Arc where NPY/AgRP- or α-MSH-producing cells are located. More recently, we found that NPGL innervated α-MSH-producing cells and the acute i.c.v. injection of NPGL stimulated feeding behavior in mice fed with normal chow (*Matsuura et al., 2017*). Taken together, NPGL may regulate the transcription or activity of these well-known orexigenic or anorexigenic factors to modulate energy homeostasis involved in food intake and fat accumulation in rats. In the present study, chronic i.c.v. infusion of NPGL did not increase food intake in rats fed normal chow (*Figure 4E*), whereas a single injection of a lower dose of NPGL has been shown to increase food intake in mice (*Matsuura et al., 2017*). The reason for this discrepancy is unclear and future studies in which rats and mice are investigated under identical conditions are needed to clarify whether differences exist between species in the role played by this neuropeptide. Furthermore, the VTM is known as the origin of histaminergic neurons, and neuronal histamine is also an anorexigenic and anti-obesity factor (*Haas and Panula, 2003*). Thus, it is possible that NPGL-expressing cells interact with histamine neurons to regulate fat accumulation. NPGL-immunoreactive fibers were also detected around the dorsomedial hypothalamic nucleus (DMH) and

ventromedial hypothalamic nucleus (VMH) (*Figure 1D and J*). These regions also participate in energy homeostasis (*Schwartz and Porte, 2005*; *Morton et al., 2006*). In addition, it has been demonstrated that lipogenesis and lipolysis in WAT are regulated in the mediobasal hypothalamus through the sympathetic nervous system (*Buettner et al., 2008*; *Scherer et al., 2011*). Therefore, it is possible that NPGL may participate in the sympathetic regulation of adiposity. Future studies are necessary to identify the target sites and neuronal networks concerning NPGL-producing neurons.

In conclusion, the present data reveal novel molecular and functional relationships among the hypothalamus, insulin action, and peripheral adiposity and elucidate a previously unknown role for NPGL in energy homeostasis. Together, these findings reveal a novel neurochemical gateway where signals from the brain and periphery converge to monitor energetic status and adjust feeding and metabolism appropriately. The cognate receptor for NPGL has not yet been identified. The identification of the receptor will help to elucidate the mechanisms of NPGL activity in the mammalian brain. However, the present data bring to light an important role for NPGL in regulation of food intake, body mass, and onset of obesity.

## Materials and methods

### Animals

Male Wistar rats (7 weeks old) were purchased from a commercial company (Kyudo, Saga, Japan) and housed for one week at 23 ± 2°C under a 12:12 hr light–dark cycle with *ad libitum* access to tap water and normal chow (CE-2; CLEA Japan, Tokyo, Japan). In several studies, a high-fat/high-sucrose diet (HFSD; 32% of calories from fat/20% of calories from sucrose, D14050401; Research Diets, New Brunswick, NJ) was also used as a high calorie diet. In a food choice experiment, protein diet (98.5% of calories from casein/1.5% of calories from L-cystine, D14082901; Research Diets), carbohydrate diet (43.6% of calories from corn starch/40.6% of calories from sucrose/15.8% calories from maltodextrin 10, D14082902; Research Diets), and fat diet (92.2% of calories from lard/7.8% of calories from soy bean oil, D14082903; Research Diets) were used as macronutrient diets.

All animal procedures were performed in accordance with the Guide for the Care and Use of Laboratory Animals prepared by Hiroshima University (permit numbers: C11-2, C13-12, and C13-17).

### Production of NPGL and antibodies against NPGL

Rat NPGL of 80 amino acid residues, its analogs NPGL-Gly and NPGL(32-80), and NPGM (*Figure 1—figure supplement 1E*) were synthesized by microwave-assisted solid-phase peptide synthesis using an automated peptide synthesizer (Syro Wave; Biotage, Uppsala, Sweden) as previously described (*Masuda et al., 2015*).

Rabbit antisera were produced following our published method (*Ukena et al., 2010*) using synthetic NPGL as the antigen. Antigen solution was mixed with Freund's complete adjuvant and injected into rabbits. After a booster injection, blood was collected from each rabbit and the optimal serum with high titer was selected by a dot-blot analysis (*Figure 1—figure supplement 1F*). The rabbit antibody against NPGL (RRID: AB_2636993) was purified on an NPGL-conjugated sepharose 6B column. The guinea pig antibody against NPGL (RRID: AB_2636992) was similarly produced.

### Production of AAV-based vectors

We followed a previously reported method (*Chen et al., 2007*). The full-length open reading frame of rat NPGL was amplified from cDNA of the MBH and inserted into pAAV-IRES-GFP expression vector (Cell Biolabs, San Diego, CA). The primers for rat NPGL were 5′-CGCGGATCCACCATGACTGATCCTGGGA-3′ for sense primer and 5′-CGGAATTCTTAGCTTCGATTTCTCTTTATT-3′ for antisense primer.

AAV-based vectors AAV-DJ/8-NPGL-IRES-GFP for NPGL (AAV-NPGL) and AAV-DJ/8-IRES-GFP for control (AAV-CTL) were produced in 293AAV cells (Cat# AAV-100; Cell Biolabs) using AAV-DJ/8 Helper Free Packaging System containing pAAV-DJ/8 and pHelper plasmids (Cell Biolabs). The triple plasmids (AAV-DJ/8-NPGL-IRES-GFP or AAV-DJ/8-IRES-GFP, pAAV-DJ/8, and pHelper) were mixed with the Polyethylenimine MAX transfection reagent (PEI-MAX; Polysciences, Warrington, PA); the mixture was diluted with Opti-MEM I medium (Life Technologies, Carlsbad, CA) and added to

293AAV cells in 150 mm dishes. Transfected cells were cultured in DMEM containing 10% fetal bovine serum.

For the purification of AAV-based vectors, three days after transfection, the cells and supernatants were harvested and purified using chloroform and condensed using Amicon Ultra-4 Centrifugal Filter Devices (100K MWCO; Merck Millipore, Billerica, MA).

For AAV titration, 1 μl of AAV solution was treated with RQ1 DNase (Promega, Madison, WI) according to manufacturer's directions. Virus titers were determined by quantitative PCR with EGFP primer pairs. The primers for EGFP were 5'-ACCACTACCTGAGCACCCAGTC-3' for sense primer and 5'-GTCCATGCCGAGAGTGATCC-3' for antisense primer. After titration, the AAV-based vectors were prepared at a concentration of $1 \times 10^9$ particles/μl and stored at −80°C until use.

## Stereotaxic surgery

The stereotaxic coordinates used in the surgery were plotted according to the Rat Brain Atlas of Paxinos and Watson (*Paxinos and Watson, 2007*). Eight-week-old rats weighing 270–310 g were placed in a stereotaxic frame (model 963; David Kopf Instruments, Tujunga, CA) under isoflurane anesthesia.

### Overexpression

For *Npgl* overexpression, rats were bilaterally injected with 1 μl/site ($1 \times 10^9$ particles/site) of AAV-based vectors (AAV-NPGL or AAV-CTL) using a Neuros Syringe (7001KH; Hamilton) into the medio-basal hypothalamic region with the coordinates 3.8 mm caudal to bregma, 0.6 mm lateral to midline, and 9.8 mm ventral to skull surface. *Npgl* overexpression was kept for six weeks with normal chow or twelve weeks with a high calorie diet. *Npgl* overexpression was confirmed by real-time RT-PCR at the endpoint. Food intake and body weight were measured every day. The masses of WAT, BAT, liver, heart, kidney and muscle were measured at the endpoint. Body length was measured under isoflurane anesthesia at forty-two days after injecting AAV-based vectors.

### I.c.v. chronic infusion

For the 13 day chronic infusion of NPGL or the rabbit antibody against NPGL (RRID: AB_2636993), the infusion cannula (28 gauge, ALZET Brain Infusion Kit 1; DURECT Co., Cupertino, CA) was unilaterally inserted into the left lateral ventricle. The final coordinates of the cannula tips were as follows: 0.9 mm caudal to bregma, 1.5 mm lateral to midline, and 4.5 mm ventral to the skull surface.

NPGL (7.5 or 15 nmol/day) was dissolved in 30% propylene glycol and adjusted to pH 8.0 with NaOH. The dose was determined on the basis of the previous study of galanin-like peptide (*Rich et al., 2007*). For the control, the vehicle solution was employed.

The rabbit antibody against NPGL (400 ng/day) was dissolved in 10 mM phosphate-buffered saline (PBS). The dose was based on the previous report of immunoneutralization of interleukin-6 (*Donegan et al., 2014*). For the control, normal rabbit IgG (400 ng/day) was infused.

The solutions were loaded into an ALZET mini-osmotic pump (delivery rate 0.5 μl/hr, model 2002 for two weeks; DURECT Co.) connected with the infusion cannula using polyethylene tubing a day before surgery and kept overnight at 37°C. On the day of cannula insertion, the osmotic pump was implanted subcutaneously into the back. We confirmed that the correct infusion occurred by examining the solution remaining in the pump at the endpoint. In addition, it was also confirmed that the remaining antibody against NPGL in the pump binds NPGL by a dot-blot analysis.

Food intake and body mass were measured every day. The mass of WAT was measured at the endpoint on all experiments, and the masses of BAT, liver, heart, kidney and muscle were also measured at the endpoint on the experiment under macronutrient diets.

### I.c.v. acute injection

For the acute injection of insulin, a guide cannula (22 gauge, C313GA; Plastics One, Roanoke, VA) was implanted unilaterally into the left lateral ventricle. The cannula was fixed to the skull using acrylic resin (Shofu, Kyoto, Japan). The final coordinates of the cannula tips were 0.9 mm caudal to bregma, 1.5 mm lateral to midline, and 3.5 mm ventral to the skull surface. The injector (28 gauge, C313LI; Plastics One) extended 1.0 mm below the tips of the guide cannula. Injections were

**Table 3.** Sequences of oligonucleotide primers for real-time RT-PCR.

**A. Sequences of oligonucleotide primers for real-time RT-PCR in human tissues.**

| Gene | Forward primer | Reverse primer | Accession no. |
|---|---|---|---|
| NPGL | GGAACCATGGCTTAGGAAGG | CCTTAGGAGCTGAGAATATGTA | NM_001102659.1 |
| ACTB | GGCACCACACCTTCTACAAT | AGGTCTCAAACATGATCTGG | NM_001101.3 |

**B. Sequences of oligonucleotide primers for real-time RT-PCR in rat tissues.**

| Gene | Forward primer | Reverse primer | Accession no. |
|---|---|---|---|
| Npgl | GGAACCATGGCTTAGGAAGG | TCTAAGGAGCTGAGAATATGCA | LC003309 |
| Acc | GAGGTGGATCAGAGATTTCA | TTCAGCTCTAACTGGAAAGC | NM_022193.1 |
| Fas | AGGATGTCAACAAGCCCAAG | ACAGAGGAGAAGGCCACAAA | NM_017332.1 |
| Scd1 | TGAAAGCTGAGAAGCTGGTG | CAGTGTGGGCAGGATGAAG | NM_139192.2 |
| Gpat1 | CAGCGTGATTGCTACCTGAA | CGGAAGGTGTGGACAAAGAT | NM_017274.1 |
| Adpn | GGGCTACGCTATGTCTGAGC | GAGACTGCACACGAAGGTGA | NM_001282324.1 |
| Cpt1a | GGATGGCATGTGGGTAAAAG | TACTGACACAGGCAGCCAAA | NM_031559.2 |
| Atgl | GCTGCAAGTGGGTTTTTGAT | GTGAACGGTAAGGCACAGGT | NM_001108509.2 |
| Hsl | GAGACGGAGGACCATTTTGA | CGGAGGTCTCTGAGGAACAG | NM_012859.1 |
| Srebp1 | TCACAGATCCAGCAGGTCCCC | GGTCCCTCCACTCACCAGGGT | NM_001276707.1 |
| Lpl | TCTCCTGATGATGCGGATTT | CAACATGCCCTACTGGTTTC | NM_012598.2 |
| Cd36 | GAGGTCCTTACACATACAGAGTTCGTT | ACAGACAGTGAAGGCTCAAAGATG | NM_031561.2 |
| Fatp1 | GCGGCGTTCGGTGTGTAC | GCACGCGGATCAGAACAGA | NM_053580.2 |
| Slc2a2 | GACATCGGTGTGATCAATGC | TGTCGTATGTGCTGGTGTGA | NM_012879 |
| Slc2a4 | CCTCCAGGATGAAGGAAACA | GGGAGAAAAGCCCATCTAGG | NM_012751 |
| Gapdh | CGGCAAGTTCAACGGCACAG | ACTCCACGACATACTCAGCAC | NM_017008.4 |
| Gk | TTGAGACCCGTTTCGTGTCA | AGGGTCGAAGCCCCAGAGT | NM_001270850.1 |
| L-Pk | TGATGATTGGACGCTGCAA | GAGTTGGTCGAGCCTTAGTGATC | NM_012624.3 |
| Npy | TATCCCTGCTCGTGTT | GATTGATGTAGTGTCGCAGA | NM_012614.2 |
| Agrp | GCAGACCGAGCAGAAGATGT | GACTCGTGCAGCCTTACACA | NM_033650.1 |
| Pomc | TAAGAGAGGCCACTGAACAT | GTCTATGGAGGTCTGAAGCA | NM_139326.2 |
| Actb | GGCACCACACTTTCTACAAT | AGGTCTCAAACATGATCTGG | NM_031144.3 |
| Rps18 | AAGTTTCAGCACATCCTGCGAGTA | TTGGTGAGGTCAATGTCTGCTTTC | NM_213557.1 |

**C. Sequences of oligonucleotide primers for real-time RT-PCR in mouse tissues.**

| Gene | Forward primer | Reverse primer | Accession no. |
|---|---|---|---|
| Npgl | GGAACCATGGCTTAGGAAGG | TCTAAGGAGCTGAGAATATGCA | LC088498 |
| Actb | GGCACCACACCTTCTACAAT | AGGTCTCAAACATGATCTGG | NM_007393.4 |

delivered using a syringe (Hamilton, Reno, NV) connected by polyethylene tubing to the microinjector. The injection of reagents was done after a postoperative period of 10 days.

NPGL (5 nmol/5 μl) was dissolved in 30% propylene glycol and adjusted to pH 8.0 with NaOH. For the control, the vehicle solution was employed. rWAT and liver were collected after 5 hr of i.c.v. injection of vehicle or NPGL.

Regular human insulin (humulin R; Eli Lilly Japan, Hyogo, Japan) was diluted in saline. The injection doses were 10 mU/animal. Saline was used as a control. Forty-five min after insulin injection, rats were transcardially perfused with 4% paraformaldehyde (PFA) solution.

## Real-time RT-PCR

To survey *Npgl* mRNA expression in human, rat, and mouse tissues, multiple tissue cDNA panels were employed according to the manufacturer's directions (Clontech Laboratories, Mountain View, CA). In addition, the whole brain, telencephalon, diencephalon, mesencephalon, cerebellum, MBH, rWAT, and liver were dissected from rats and snap frozen in liquid nitrogen for RNA processing at the endpoint of NPGL infusion and *Npgl* overexpression. RNA was extracted using TRIzol reagent for brain tissues and liver (Life Technologies) or QIAzol lysis reagent for adipose tissues (QIAGEN, Venlo, Netherlands) following the manufacturer's instructions. First-strand cDNA was synthesized from total RNA using a ReverTra Ace kit (TOYOBO, Osaka, Japan).

The sequences of primers used in this study are listed in *Table 3*. PCR amplifications were conducted with THUNDERBIRD SYBR qPCR Mix (TOYOBO) using the following conditions: 95℃ for 20 s, followed by 40 cycles of 95℃ for 3 s, and 60℃ for 30 s. The PCR products in each cycle were monitored using a StepOne Real-Time Thermal Cycler (Life Technologies). Relative quantification of each gene was determined by the $2^{-\Delta\Delta Ct}$ method using $\beta$-actin (*Actb*) for brain tissues and liver or ribosomal protein S18 (*Rps18*) for rWAT as internal controls.

## Sampling procedures of MBH

*Npgl* mRNA levels in several conditions were measured. For food deprivation, MBH was harvested from forty-eight hours fasted rats and frozen down immediately in liquid nitrogen. The control rats were fed with *ad libitum*. For induction of experimental diabetes, rats were injected i.p. with streptozotocin (50 mg/kg; Sigma–Aldrich, St. Louis, MO). One week after injection when blood glucose levels were higher in the streptozotocin-injected rats, MBH was collected. For injection of insulin, rats were injected i.p. with regular human insulin (3 U/kg, humulin R) after overnight fasting. Seven hours after injection, MBH were harvested. Blood glucose levels were lower in the insulin-injected rats at this time. *Npgl* mRNA expression levels in MBH were measured by real-time RT-PCR.

## Western blot analysis

For the detection of mature NPGL in the hypothalamus, the hypothalami from five rats were boiled and homogenized in 5% acetic acid. The homogenate was centrifuged at 10,000 × g for 20 min at 4℃ and the precipitate was collected. After extraction with dimethyl sulfoxide, the supernatant was passed through a disposable C18 cartridge column (Sep-Pak Vac; Waters, Milford, MA). The retained material was subjected to reversed-phase HPLC using a C4 column (Protein C4-300, 4.6 × 150 mm; Tosoh, Tokyo, Japan) with a linear gradient of 35–75% acetonitrile containing 0.1% trifluoroacetic acid for more than 80 min at a flow rate of 0.5 ml/min. The fractions were evaporated, dissolved in SDS sample buffer, and subjected to 15% SDS-PAGE. After transfer onto polyvinylidene fluoride membrane (Immobilon-P; Merck Millipore), the blot was probed with the rabbit antibody against NPGL (RRID: AB_2636993, 1:1000 dilution) and incubated with horseradish peroxidase-labeled donkey anti-rabbit IgG (RRID: AB_772206, 1:1000 dilution, Cat# NA934; GE Healthcare, Little Chalfont, England). The protein bands were detected by ECL Prime (GE Healthcare) or ECL Pro (PerkinElmer, Waltham, MA) western blotting detection reagents.

For the detections of FAS, phospho-HSL (pHSL) and total HSL in rWAT, the tissue was homogenized in ice-cold RIPA buffer (Thermo Fisher Scientific, Waltham, MA), Halt Protease Inhibitor Cocktail, EDTA-Free (Thermo Fisher Scientific) and centrifuged at 10,000 × g for 20 min to remove the lipids. Protein content in the aqueous solution was measured using the Pierce BCA Protein Assay Kit (Thermo Fisher Scientific) with BSA as a standard, 30 µg-aliquots were subjected to 6% SDS-PAGE. The western blot procedure was similar to the one described above. The antibodies against FAS, pHSL, total HSL, and α-tubulin as an internal control were employed for FAS (RRID: AB_2100801, 1:1000 dilution, Cat# 10624–2-AP; Proteintech, Chicago, IL), pHSL (RRID: AB_490997, 1:500 dilution, Cat# 4126; Cell Signaling Technology, Danvers, MA), total HSL (RRID: AB_2296900, 1:500 dilution, Cat# 4107; Cell Signaling Technology), and for α-tubulin (RRID: AB_10598496, 1:2000 dilution, Cat# PM054; Medical and Biological Laboratories, Nagoya, Japan).

## Histological analysis

### In situ hybridization

Rats were transcardially perfused with 4% PFA solution and the brains were post-fixed overnight at 4°C and then put in a refrigerated sucrose solution (30% sucrose in 10 mM phosphate buffer) until they sank. The brain tissues were embedded in Tissue-Tek O.C.T. Compound (Sakura Finetek, Tokyo, Japan), sectioned coronally to a thickness of 16 µm with a cryostat at −20°C, and then mounted on slides. The in situ hybridization procedure was similar to that previously described (*Ukena et al., 1999*, *2008*).

Digoxigenin (DIG)-labeled antisense and sense RNA probes were used. The DIG-labeled RNA probes were produced from a portion of the NPGL precursor cDNA using the DIG-RNA labeling kit [SP6/T7] (Roche Diagnostics, Basel, Switzerland). The DIG-labeled sense RNA probe, which was complementary to a sequence of the antisense probe, was used as a control to verify specificity.

The brain sections were rehydrated with PBS and treated with 0.2 N HCl for 20 min, followed by incubation with 1 µg/ml proteinase K at 37°C for 10 min. After fixation with 4% PFA in PBS for 5 min, the slides were incubated in 40% deionized formamide in 4 × SSC (1 × SSC = 150 mM NaCl and 15 mM sodium citrate, pH 7.0) for 30 min. Hybridization was performed at 50°C overnight with the DIG-RNA probe mixture dissolved in the hybridization medium containing 10 mM Tris-HCl (pH 7.4), 1 mM EDTA, 0.6 M NaCl, 10% dextran sulfate, 1× Denhardt's solution, 250 µg/ml yeast tRNA, 125 µg/ml salmon sperm DNA, and 40% deionized formamide. The sections were washed six times with 50% formamide/2 × SSC at 55°C for 10 min and treated with blocking solution (1% BSA, 1% normal goat serum, 0.3% Triton-X100 in PBS). They were then incubated with alkaline phosphatase-labeled rabbit anti-DIG antibody (RRID: AB_514497, 1:1000 dilution, Cat# 11093274910; Roche Diagnostics) for 1 hr. Subsequently, the sections were washed three times for 10 min in 0.075% Brij35 in PBS. Signals were detected after the immersion of the sections overnight in NBT/BCIP stock solution (1:50 dilution in alkaline buffer; 0.1 M Tris-HCl, pH 9.5, 0.1 M NaCl, 50 mM MgCl$_2$) under a microscope (Eclipse E600; Nikon, Tokyo, Japan).

### Immunohistochemistry

The brain tissues were fixed as described above and then sectioned into 30 µm slices for the free-floating method as previously described (*Ukena et al., 1998*). In some cases, colchicine injections (100 µg, i.c.v.) were administered two days before sacrifice.

For the immunohistochemical staining of NPGL, endogenous peroxidase activity was eliminated from the sections by incubation with 3% H$_2$O$_2$ in absolute methanol for 30 min. After blocking nonspecific binding components with 1% normal goat serum and 1% BSA in PBS containing 0.3% Triton X-100 for 1 hr at room temperature, the sections were incubated with the rabbit antibody against NPGL (RRID: AB_2636993, 1:1000 dilution) overnight at 4°C. The primary immunoreaction was executed by incubation with biotinylated goat anti-rabbit IgG (RRID: AB_2313606, 1:1000 dilution, Cat# BA-1000; Vector Laboratories, Burlingame, CA). Immunoreactive products were detected with an ABC kit (VECTASTAIN Elite Kit; Vector Laboratories), followed by diaminobenzidine reaction. Specificity control was obtained by preadsorbing the working dilution of the primary antibody with a saturating concentration (10 µg/ml) of NPGL or its paralogous protein, NPGM (*Figure 1—figure supplement 1G−L*). The localization of immunoreactive cells was studied using a microscope.

For the double-immunohistochemical staining of NPGL/phospho-Akt, permeabilization was performed by methanol for 30 min. After blocking nonspecific binding components with 1% normal donkey serum and 1% BSA in PBS containing 0.3% Triton X-100 for 1 hr at room temperature, the sections were incubated with guinea pig antibody against NPGL (RRID: AB_2636992, 1:200 dilution)/rabbit antibody against phospho-Akt (RRID: AB_2315049, 1:400 dilution, Cat# 4060; Cell Signaling Technology, Danvers, MA) overnight at 4°C. Alexa Fluor 488-conjugated donkey anti-guinea pig IgG (H+L) (RRID: AB_2340472, 1:600 dilution, Cat# 706-545-148; Jackson ImmunoResearch, West Grove, PA) and Cy3-conjugated donkey anti-rabbit IgG (RRID: AB_2307443, 1:400 dilution, Cat# 711-165-152; Jackson ImmunoResearch) were used as a secondary antibody.

### Hematoxylin and eosin staining

rWAT was soaked in 4% PFA at the endpoint of NPGL infusion and *Npgl* overexpression, embedded in Neg-50 Frozen Section Medium (Thermo Fisher Scientific), and sectioned to a thickness of 20 µm

with a cryostat at −30℃. The sections were then air-dried and delipidated with acetone. The nucleus and cytoplasm were stained with hematoxylin and eosin (5 min for each stain), and the sections were washed in tap water. After dehydration with alcohol series and clearing with xylene, the sections were mounted on slides and examined under a microscope.

### Measurement of adipocyte size

The slides were photographed under a light microscope. For rWAT analysis, two regions were randomly selected from each slide and diameters of approximately 100–250 adipocytes were obtained by averaging the largest and smallest diameters. Two independent investigators who were blind to the groups performed all analyses.

## Indirect calorimetry and locomotor activity

Four weeks after injecting AAV-based vectors, indirect calorimetry was performed using an $O_2/CO_2$ metabolism-measuring system for small animals (MK-5000RQ; Muromachi Kikai, Tokyo, Japan). The system monitored $VO_2$ (ml/min) and $VCO_2$ (ml/min) at 3 min intervals and calculated the respiratory quotient (RQ) ratio ($VCO_2/VO_2$). Locomotor activity was simultaneously measured using the Super-Mex infrared ray passive sensor system (Muromachi Kikai). Measurements were collected hourly over a 23 hr period (light period: 10:00–21:00, dark period: 21:00–9:00) after habituation for 30 min. Energy expenditure was calculated using the following equation: energy expenditure (cal/kg/hr) = $VO_2$ (ml/kg/hr) $\times$ [3.815 + (1.232 $\times$ RQ)] (*Lusk, 1928*).

## Fatty acid analysis

For the analysis of endogenous SCD1 activity in rWAT, the lipids were extracted according to the previous method (*Folch et al., 1957*). WAT (100 mg) was extracted with 1 ml of chloroform: methanol (2:1) using beads crusher (μT-12; TAITEC, Saitama, Japan) and 0.25 ml of distilled water was added and mixed by inversion. After incubation for 30 min, the sample was centrifuged at 3000 $\times$ g and the lower organic phase was collected and evaporated. Extracted fatty acids were methylated using Fatty Acid Methylation Kit (nacalai tesque, Kyoto, Japan) and purified using Fatty Acid Methyl Ester Purification Kit (nacalai tesque). The eluted solution was evaporated to dryness and kept at –20℃. The residues were resolved into hexane and fatty acids were identified by GC-MS (JMS-T100 GCV; JEOL, Tokyo, Japan). The SCD1 activity was estimated as oleate to stearate ratio (18:1/18:0) and palmitoleate to palmitate ratio (16:1/16:0) from individual fatty acids. The 16:1/16:0 ratio seems to be a better indicator of endogenous SCD1 activity than the 18:1/18:0 ratio (*Sampath and Ntambi, 2008*).

## Blood tests

Serum levels of glucose, lipids, and hormones were measured using appropriate equipment, reagents, and kits. The GLUCOCARD G+ meter was used to measure glucose content (Arkray, Kyoto, Japan). The NEFA C-Test Wako (Wako Pure Chemical Industries, Osaka, Japan) was used to measure free fatty acid levels. The Triglyceride E-Test Wako (Wako Pure Chemical Industries) was used to measure triglyceride levels and the Cholesterol E-Test Wako (Wako Pure Chemical Industries) for cholesterol content. The Rebis Insulin-rat T ELISA kit (Shibayagi, Gunma, Japan) was used to measure insulin levels, the Leptin ELISA kit (Morinaga Institute of Biological Science, Yokohama, Japan) for leptin, and the Corticosterone ELISA kit (Assaypro, St. Charles, MO) for corticosterone. Blood chemistry is listed in *Tables 1* and *2*.

## Statistical analysis

Data were analyzed using the student's *t*-test or two-way analysis of variance (ANOVA) followed by Bonferroni's test for two groups. In addition, data from three or six groups were statistically analyzed using one-way ANOVA with Tukey's post-hoc test as appropriate. Statistical significance was set at $p < 0.05$. All results are presented as the mean ± standard error of the mean (± s.e.m.).

## Acknowledgements

We thank Yukie Tanaka, Haruka Ooyama, Yoshihiro Ohguchi, Keiko Masuda, Yuki Bessho, Sho Maejima, Daichi Matsuura, Masaki Kato, and Takaya Saito for experimental support; Drs. Yojiro Muneoka, Yumiko Saito, Yasuo Furukawa, Tetsuya Tachibana, Noboru Murakami, Yukari Date, Sayaka Akieda-Asai, and Takanori Ida, for discussion. This work was supported by grants from Japan Society for the Promotion of Science KAKENHI Grants (JP22687004, JP23126517, JP25126717, JP26291066, and JP15KK0259 to KU and JP25440171 to EI-U), Grant-in-Aid for Japan Society for the Promotion of Science Fellows (15J03781 to KS), the Program for Promotion of Basic and Applied Research for Innovations in Bio-oriented Industry (KU), the Toray Science Foundation (KU), the Mishima Kaiun Memorial Foundation (KU and EI-U), the Suzuken Memorial Foundation (KU), the SKYLARK Food Science Institute (KU), the Urakami Foundation for Food and Food Culture Promotion (KU), and the Kao Research Council for the Study of Healthcare Science (KU).

## Additional information

### Funding

| Funder | Grant reference number | Author |
|---|---|---|
| Japan Society for the Promotion of Science | KAKENHI JP22687004 | Kazuyoshi Ukena |
| Japan Society for the Promotion of Science | KAKENHI JP23126517 | Kazuyoshi Ukena |
| Japan Society for the Promotion of Science | KAKENHI JP25126717 | Kazuyoshi Ukena |
| Japan Society for the Promotion of Science | KAKENHI JP26291066 | Kazuyoshi Ukena |
| Japan Society for the Promotion of Science | KAKENHI JP15KK0259 | Kazuyoshi Ukena |
| Japan Society for the Promotion of Science | KAKENHI JP25440171 | Eiko Iwakoshi-Ukena |
| Grant-in-Aid for Japan Society for the Promotion of Science Fellows | 15J03781 | Kenshiro Shikano |
| Bio-oriented Technology Research Advancement Institution | | Kazuyoshi Ukena |
| Toray Science Foundation | | Kazuyoshi Ukena |
| Mishima Kaiun Memorial Foundation | | Eiko Iwakoshi-Ukena Kazuyoshi Ukena |
| Suzuken Memorial Foundation | | Kazuyoshi Ukena |
| Skylark Food Science Institute | | Kazuyoshi Ukena |
| Urakami Foundation for Food and Food Culture Promotion | | Kazuyoshi Ukena |
| Kao Research Council for the Study of Healthcare Science | | Kazuyoshi Ukena |

The funders had no role in study design, data collection and interpretation, or the decision to submit the work for publication.

### Author contributions

EI-U, Conceptualization, Data curation, Funding acquisition, Investigation, Writing—original draft, Writing—review and editing; KS, Data curation, Formal analysis, Funding acquisition, Investigation, Writing—original draft, Writing—review and editing; KK, ST, MF, YO, Data curation, Investigation; TS, SO, GEB, LJK, YM, Validation, Writing—review and editing, Assist experimental design; KU,

Conceptualization, Formal analysis, Supervision, Funding acquisition, Investigation, Writing—original draft, Writing—review and editing

## Author ORCIDs
Kenshiro Shikano, http://orcid.org/0000-0003-4395-3787
Kazuyoshi Ukena, http://orcid.org/0000-0003-0942-6035

## Ethics

Animal experimentation: All animal procedures were performed in accordance with the Guide for the Care and Use of Laboratory Animals prepared by Hiroshima University (permit numbers: C11-2, C13-12, and C13-17).

# Additional files

## Supplementary files

• Supplementary file 1. The detailed statistical analysis of figures. The table shows the statistics test, number of n, comparison, p value, and degree of freedom and F/t/z/ETC value in each figure panel.

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
