## [Decision Letter]

Thank you for submitting your article "Neurosecretory protein GL stimulates food intake, de novo lipogenesis, and onset of obesity" for consideration by *eLife*. Your article has been favorably evaluated by a Senior Editor and three reviewers, one of whom, Richard D Palmiter (Reviewer #1), is a member of our Board of Reviewing Editors. The following individuals involved in review of your submission have agreed to reveal their identity: Serge Luquet (Reviewer #2); Don Marsh (Reviewer #3).

The reviewers have discussed the reviews with one another and the Reviewing Editor has drafted this decision to help you prepare a revised submission.

Summary:

The authors of this paper provide initial characterization of a newly discovered neuropeptide that they call NPGL. It is expressed in a small number of neurons in the arcuate region of the hypothalamus and is modestly regulated by diet and insulin. Chronic expression (from a virus) or infusion (icv) of NPGL has a modest effect on food intake and body weight when rats are fed a high-fat diet but has minimal effect when fed normal chow. However, even without large effects on food intake there are significant effects on adipocyte physiology. The authors suggest that this neuropeptide plays a role in fat storage during times of energy restriction and contributes to obesity when over-expressed in animals on a high-fat diet.

The findings described in this manuscript are novel and will be of significant interest to scientists and physicians studying/working in the field of metabolic disorders. We have no major concerns with this paper and support its publication as an original article in *eLife*.

Reviewer #1 (Minor Comments):

Introduction, first paragraph – most of these general references are more than 10 years out of date.

Introduction, second paragraph – when referring to genes, gene names should be in italics. Has the *Npgl* gene name been approved by Mouse Genome Informatics?

Introduction, second paragraph – rat NPGL has a di-sulfide, but is it really circular?

Subsection “Effects of NPGL-precursor gene overexpression on food intake, body mass, adiposity, and blood insulin”, first paragraph – it would be better to say that we prepared an AAV that would allow expression of the NPGL precursor protein (Figure 2—figure supplement 1) and injected it into the hypothalamus of rats. We monitored food intake (normal chow) for 6 weeks (Figure 2). It would be better to spell out OE rather than abbreviate it.

Subsection “Effects of NPGL-precursor gene overexpression on food intake, body mass, adiposity, and blood insulin”, first paragraph – did the mass of the muscles actually decrease or not increase as much as controls? If it decreased that implies muscle wasting.

Subsection “Effects of NPGL-precursor gene overexpression on food intake, body mass, adiposity, and blood insulin”, last paragraph – I would emphasize the positive results with the high-fat diet first, and then mention the minimal effects with normal chow second (or in supplemental).

Figure 3, Figure 4 put mRNA names in italics.

Subsection “Effect of i.c.v. infusion of antibody against NPGL on lipid droplets of WAT” – what is the evidence that the antibody is actually effective at blocking NPGL action?

Subsection “Effects of NPGL on selective carbohydrate intake, adiposity, blood insulin, and de novo lipogenesis”, last paragraph – only use approved mouse gene names in italics: *Glut2* should be *Slc2a*2 and *Glut4* should be *Slc2a4*.

Subsection “Response of NPGL to fasting and insulin” and Discussion, fourth paragraph – "expression" not expressions.

Subsection “Response of NPGL to fasting and insulin” – this paragraph is difficult to understand. Authors should elaborate more on the models chosen and the results.

Discussion, first paragraph – have the authors read this *eLife* paper: Lanfray et al. *eLife* 2016;5:e11742. DOI: 10.7554/*eLife*.11742?

The first paragraph of Discussion usually makes a general conclusion from the paper. This first part of the paragraph belongs in Introduction and the second half mentions other peptides without ascribing any function to them. How does NPGL fit into the overall scheme?

Discussion, second paragraph – "remarkably" relative to what?

Discussion, second paragraph – no em dash and hyphen for "finally".

Discussion, fifth paragraph – what is meant by "descending" in this context?

It is curious that infusion of 15 nmol of NPGL/day had no effect on food intake in rats fed a normal diet, but injection of 1 nmol/mouse had a significant effect on food intake (Matsuura, 2017). What is the explanation for this discrepancy?

Reviewer #2 (Minor Comments):

1) When% of tissue weight is provided it would be important also to have raw data as it could indicate that change in mass contribution (for soleus for instance) is not due to muscle mass loss but rather body weight change.

2) As part of a potential mechanism by which central NPGL controls peripheral insulin release, sensitivity and carbohydrate/lipid balance the authors should comment on the potential role for the autonomic nervous system as it obviously seems to be a likely mediator of the effect.

Reviewer #3 (Minor Comments):

At several places in the Results section and in the headers of Table 1 and Table 2, the authors refer to the measurements of insulin and leptin as 'blood composition' or 'blood components'. I would encourage the authors to consider changing this to 'blood chemistry' since blood components typically refer to blood cell types and coagulation factors.

---

## [Author Response]

Reviewer #1 (Minor Comments):

Introduction, first paragraph – most of these general references are more than 10 years out of date.

Based on this comment, we have added more recent references in the Introduction.

Introduction, second paragraph – when referring to genes, gene names should be in italics. Has the Npgl gene name been approved by Mouse Genome Informatics?

Based on this comment, gene names are now in italics.

As we have deposited the cDNA sequence as “precursor of NPGL” in the GenBank database recently, *Npgl* should be approved by Mouse Genome Informatics soon.

Introduction, second paragraph – rat NPGL has a di-sulfide, but is it really circular?

Based on this comment, we have changed the text as follows: “Rat NPGL assumes a circular structure, although the mature structure has not been determined (Figure 1).”

In the present study, we have revealed that the longer form containing a disulfide bond is functional by the analysis of the structure-activity relationship (Figure 4—figure supplement 3 and subsection “Effects of i.c.v. infusion of NPGL on food intake, adiposity, and *de novo* lipogenesis”, last paragraph).

Subsection “Effects of NPGL-precursor gene overexpression on food intake, body mass, adiposity, and blood insulin”, first paragraph – it would be better to say that we prepared an AAV that would allow expression of the NPGL precursor protein (Figure 2—figure supplement 1) and injected it into the hypothalamus of rats. We monitored food intake (normal chow) for 6 weeks (Figure 2). It would be better to spell out OE rather than abbreviate it.

Based on this comment, we have changed the texts as follows: “we prepared an adeno-associated virus (AAV) that would allow chronic expression of the NPGL precursor protein (Figure 2—figure supplement 1) and injected it into the hypothalamus of rats (Figure 2—figure supplement 1). We then monitored food intake (high-fat/high-sucrose diet; high calorie diet, and normal chow) for 6 weeks (Figure 2).”

In addition, we have changed “OE” to “overexpression” throughout the manuscript.

Subsection “Effects of NPGL-precursor gene overexpression on food intake, body mass, adiposity, and blood insulin”, first paragraph – did the mass of the muscles actually decrease or not increase as much as controls? If it decreased that implies muscle wasting.

Based on this comment, we have changed the text as follows: “while the masses of soleus and gastrocnemius muscles did not increase as much as controls”.

In addition, we have added the raw data (g) in Figure 2—figure supplement 2 and Figure 5.

Subsection “Effects of NPGL-precursor gene overexpression on food intake, body mass, adiposity, and blood insulin”, last paragraph – I would emphasize the positive results with the high-fat diet first, and then mention the minimal effects with normal chow second (or in supplemental).

Based on this comment, we have mentioned the positive results with the high-fat diet first, and then mentioned the minimal effects with normal chow second in both experiments of gene overexpression and protein infusion.

Figure 3, Figure 4 put mRNA names in italics.

Based on this comment, we have put mRNA names in italics in Figure 3 and 4.

Subsection “Effect of i.c.v. infusion of antibody against NPGL on lipid droplets of WAT” – what is the evidence that the antibody is actually effective at blocking NPGL action?

Based on this comment, we have added the text as follows: “In addition, it was also confirmed that the remaining antibody against NPGL in the pump binds NPGL by a dot-blot analysis” in the Materials and methods.

*Subsection “Effects of NPGL on selective carbohydrate intake, adiposity, blood insulin, and* de novo *lipogenesis”, last paragraph – only use approved mouse gene names in italics: Glut2 should be Slc2a2 and Glut4 should be Slc2a4.*

Based on this comment, we have changed“*Glut2*” to “*Slc2a2*”, and“*Glut4*” to “*Slc2a4*” in the last paragraph of the subsection “Effects of NPGL on selective carbohydrate intake, adiposity, blood insulin, and *de novo* lipogenesis”

Subsection “Response of NPGL to fasting and insulin” and Discussion, fourth paragraph – "expression" not expressions.

Based on this comment, we have changed “expressions” to “expression” in the subsection “Response of NPGL to fasting and insulin” and Discussion, third paragraph.

Subsection “Response of NPGL to fasting and insulin” – this paragraph is difficult to understand. Authors should elaborate more on the models chosen and the results.

Based on this comment, we have markedly modified the paragraph in the Results section titled “Response of NPGL to fasting and insulin”.

Discussion, first paragraph – have the authors read this eLife paper: Lanfray et al. eLife 2016;5:e11742. DOI: 10.7554/eLife.11742?

Based on this comment, we have added this *eLife* paper as a reference.

The first paragraph of Discussion usually makes a general conclusion from the paper. This first part of the paragraph belongs in Introduction and the second half mentions other peptides without ascribing any function to them. How does NPGL fit into the overall scheme?

Based on this comment, we have omitted the first paragraph in the Discussion and moved the discussion of other peptides into the Introduction (first paragraph).

Discussion, second paragraph – "remarkably" relative to what?

Based on this comment, we have omitted the word “remarkably”.

Discussion, second paragraph – no em dash and hyphen for "finally".

Based on this comment, we have omitted em dash and hyphen for "finally".

Discussion, fifth paragraph – what is meant by "descending" in this context?

Based on this comment, we have changed “the descending neuronal networks” to “the target sites and neuronal networks”.

It is curious that infusion of 15 nmol of NPGL/day had no effect on food intake in rats fed a normal diet, but injection of 1 nmol/mouse had a significant effect on food intake (Matsuura, 2017). What is the explanation for this discrepancy?

Based on this comment, we have added the texts as follows: “In the present study, chronic i.c.v. infusion of NPGL did not increase food intake in rats fed normal chow (Figure 4), whereas a single injection of a lower dose of NPGL has been shown to increase food intake in mice (Matsuura et al., 2017). The reason for this discrepancy is unclear and future studies in which rats and mice are investigated under identical conditions are needed to clarify whether differences exist between species in the role played by this neuropeptide” in the Discussion.

Reviewer #2 (Minor Comments):

1) When% of tissue weight is provided it would be important also to have raw data as it could indicate that change in mass contribution (for soleus for instance) is not due to muscle mass loss but rather body weight change.

Based on this comment, we have added the raw data (g) in Figure 2—figure supplement 2 and Figure 5.

2) As part of a potential mechanism by which central NPGL controls peripheral insulin release, sensitivity and carbohydrate/lipid balance the authors should comment on the potential role for the autonomic nervous system as it obviously seems to be a likely mediator of the effect.

Based on this comment, we have added the texts as follows: “In addition, it has been demonstrated that lipogenesis and lipolysis in WAT are regulated in the mediobasal hypothalamus through the sympathetic nervous system (Buettner et al., 2008; Scherer et al., 2011). Therefore, it is possible that NPGL may participate in the sympathetic regulation of adiposity” in the Discussion.

Reviewer #3 (Minor Comments):

At several places in the Results section and in the headers of Table 1 and Table 2, the authors refer to the measurements of insulin and leptin as 'blood composition' or 'blood components'. I would encourage the authors to consider changing this to 'blood chemistry' since blood components typically refer to blood cell types and coagulation factors.

Based on this comment, we have changed “blood composition or blood contents” to “blood chemistry” throughout the manuscript and Table 1 and 2.